# Katanin, kinesin-13, and ataxin-2 inhibit premature interaction between maternal and paternal genomes in *C. elegans* zygotes

Elizabeth A Beath, Cynthia Bailey, Meghana Mahantesh Magadam, Shuyan Qiu, Karen L McNally, Francis J McNally*

Department of Molecular and Cellular Biology, University of California, Davis, United States

*For correspondence:
fjmcnally@ucdavis.edu

Competing interest: The authors declare that no competing interests exist.

**Abstract** Fertilization occurs before the completion of oocyte meiosis in the majority of animal species and sperm contents move long distances within the zygotes of mouse and *C. elegans*. If incorporated into the meiotic spindle, paternal chromosomes could be expelled into a polar body resulting in lethal monosomy. Through live imaging of fertilization in *C. elegans*, we found that the microtubule disassembling enzymes, katanin and kinesin-13 limit long-range movement of sperm contents and that maternal ataxin-2 maintains paternal DNA and paternal mitochondria as a cohesive unit that moves together. Depletion of katanin or double depletion of kinesin-13 and ataxin-2 resulted in the capture of the sperm contents by the meiotic spindle. Thus limiting movement of sperm contents and maintaining cohesion of sperm contents within the zygote both contribute to preventing premature interaction between maternal and paternal genomes.

## eLife assessment

This is a **valuable** paper that identifies a potential challenge for embryos during fertilization: holding sperm contents in the fertilized embryos away from the oocyte meiotic spindle so that they don't get ejected into the polar body during meiotic chromosome segregation. The authors identify proteins involved in cytoplasmic streaming and maintaining the grouping of paternal organelles as being critical for this process. There remain minor weaknesses in the data presented but the paper provides **solid** evidence for the majority of its claims, and while the findings may pertain to a narrow audience the tools used and basic characterization shown will likely be relied upon by many in the community and therefore is of high value.

## Introduction

Reproduction by most animal species requires the expulsion of chromosomes into polar bodies during oocyte meiosis to reduce chromosome number and fertilization by sperm to restore a diploid chromosome number. Because fertilization occurs during oocyte meiosis in most animal species, there is an inherent risk of sperm DNA being incorporated into the meiotic spindle and being expelled into a polar body. In some species, the first line of defense against this hypothetical calamity is ensuring that sperm does not fuse with the oocyte plasma membrane directly over the meiotic spindle. A Ran-GTP gradient emanating from the mouse metaphase II meiotic spindle excludes the fusion proteins Juno and CD9 from the oocyte plasma membrane over the spindle. When this mechanism was bypassed by injecting demembranated sperm adjacent to the spindle, the sperm DNA was ejected into a polar

body (*Mori et al., 2021*). In *C. elegans*, the nucleus is positioned away from the site of future fertilization so that the meiosis I spindle assembles at the opposite end of the ellipsoid zygote from the site of fertilization (*Panzica and McNally, 2018*; *McNally et al., 2010*; *McCarter et al., 1999*).

However, simply controlling the site of fertilization is not sufficient to maintain a distance between the meiotic spindle and the sperm DNA because cytoplasmic streaming can move the sperm DNA long distances in both mouse (*Mori et al., 2021*) and *C. elegans* (*Panzica et al., 2017*; *Kimura and Kimura, 2020*). Meiotic cytoplasmic streaming in *C. elegans* embryos requires microtubules (*Yang et al., 2003*), kinesin-1 (*McNally et al., 2010*), and reticulons (*Kimura et al., 2017*), but both mechanism and purpose are not completely understood. In addition to meiotic spindle microtubules, *C. elegans* meiotic embryos have cytoplasmic microtubules around the cortex and throughout the cytoplasm of the entire embryo (*McNally et al., 2010*). These microtubules are thought to drive meiotic cytoplasmic streaming because depletion of tubulin stops cytoplasmic streaming (*Yang et al., 2003*) and depletion of the microtubule-severing protein katanin by RNAi results in an increased mass of cortical microtubules and an increase in cytoplasmic streaming (*Kimura et al., 2017*).

In addition to paternal DNA, *C. elegans* sperm introduce centrioles, SPE-11 protein, membranous organelles (MOs), and paternal mitochondria into the zygote. Paternal centrioles are silenced during meiosis by maternal KCA-1 (*McNally et al., 2012*), which is also required to pack yolk granules inward from the cortex (*McNally et al., 2010*) and the meiotic spindle outward toward the cortex (*Yang et al., 2005*). SPE-11 is an RNA-binding protein in sperm (*Li et al., 2023*) that is required paternally for polar body extrusion (*McNally and McNally, 2005*) and proper embryonic development (*Hill et al., 1989*; *Browning and Strome, 1996*). MOs are membrane vesicles in sperm that fuse with the sperm plasma membrane before fertilization during sperm activation (*L'Hernault, 2006*). However, a subset of MOs that do not fuse with the sperm plasma membrane are introduced to the zygote at fertilization and are ubiquitinated with maternal ubiquitin (*Molina et al., 2019*). Paternal mitochondria are labeled with maternal autophagy machinery (*Sato and Sato, 2011*). Whereas both MOs and paternal mitochondria are eventually destroyed during embryogenesis, during meiosis they remain in a tight cloud around the sperm DNA (*Panzica et al., 2017*; *Molina et al., 2019*; *Sato and Sato, 2011*). The mechanisms holding the sperm contents together in the zygote during cytoplasmic streaming have not been explored.

In this study, we monitored the cloud of paternal mitochondria after increasing cytoplasmic streaming or disrupting the integrity of the cloud of paternal organelles. Our results suggest that both limiting cytoplasmic streaming and maintaining the integrity of the paternal organelle cloud contribute to preventing the capture of the sperm contents by the meiotic spindle.

## Results

### Sperm-derived DNA and mitochondria are maintained in a volume that excludes maternal mitochondria and yolk granules but allows penetration by maternal ER during meiosis

Time-lapse in utero imaging of the meiotic spindle and an endoplasmic reticulum marker (ER) revealed that the ER is distributed throughout the zygote but appears as undulating lines during metaphase I (n=8), changes to a dispersed appearance during anaphase I (n=9), transitions back to undulating lines interspersed with large blobs during metaphase II (n=6), then changes to a dispersed pattern during anaphase II (n=5) (*Figure 1A*; *Figure 1—video 1*). Previous electron microscopy studies have indicated that the undulating lines correspond to sheet-like ER and the dispersed appearance corresponds to tubular ER (*Poteryaev et al., 2005*; *Gong et al., 2024*). A similar transition to sheet-like ER during mitotic M phase has been reported in HeLa cells (*Lu et al., 2009*) and *Xenopus* egg extracts (*Wang et al., 2013*), however, this may not be universal in all cell types (*Puhka et al., 2012*). Pertinent to this study, ER morphology was used to determine cell-cycle stages during live imaging reported below in Figure 5.

In striking contrast with the ER filling the entire zygote, yolk granules are packed inward, away from the cortex (n=9 metaphase I; *Figure 1B*) as previously described (*McNally et al., 2010*). When Mitotracker-treated males were mated to hermaphrodites expressing GFP-labeled yolk granules, the paternal mitochondria were found in a discrete cloud around the paternal DNA (n=34; *Figure 1B–C*) as previously described (*Sato and Sato, 2011*). Both the cloud of paternal mitochondria (n=19) and

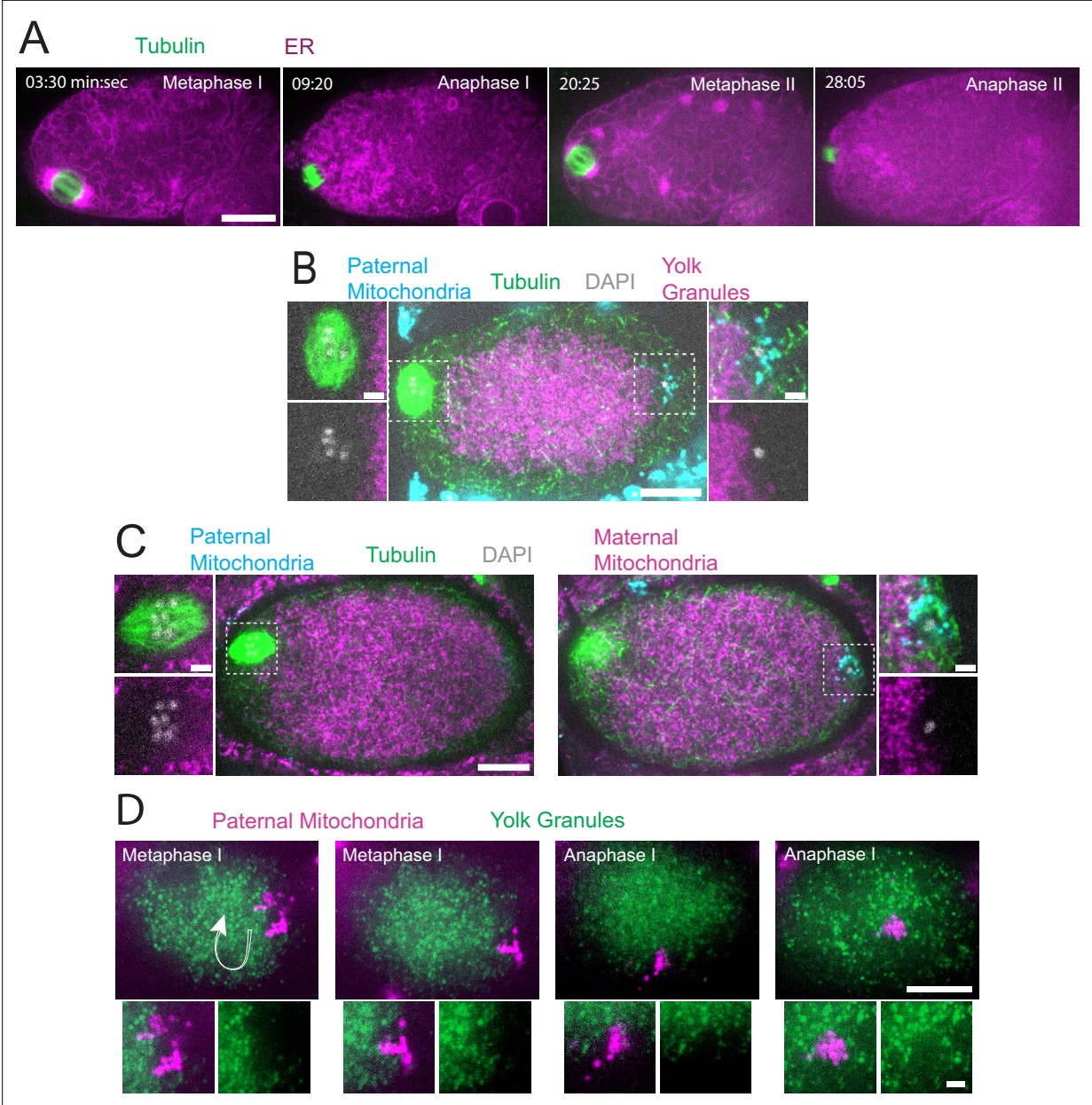

**Figure 1.** Sperm contents exclude maternal yolk granules and mitochondria. (**A**) Time-lapse in utero images of a control embryo expressing GFP::SPCS-1 (signal peptidase/ER) and mKate::TBA-2 (tubulin). ER morphology transitions from sheet-like during metaphase I and metaphase II to dispersed during anaphase I and anaphase II. (**B**) Image of a fixed metaphase I embryo expressing VIT-2::GFP (maternal yolk granules), paternal mitochondria labeled with MitoTracker Deep Red FM, and stained with alpha tubulin antibody and DAPI. n=9 metaphase I embryos with packed yolk. (**C**) Image of a fixed metaphase I embryo expressing COX-4::GFP (maternal mitochondria), paternal mitochondria labeled with MitoTracker Deep Red FM, and stained with alpha tubulin antibody and DAPI. n=5 embryos with packed maternal mitochondria. (**D**) Time-lapse in utero images of an embryo expressing VIT-2::GFP and paternal mitochondria labeled with SDHC-1::mCherry (succinate dehydrogenase). Images demonstrate sperm contents streaming in the short-axis of the embryo. Arrow indicates direction of streaming. (**A–D**) Bars: whole embryos 10 µm; insets 2 µm.

The online version of this article includes the following video for figure 1:

**Figure 1—video 1.** In utero time-lapse sequence of control embryo expressing GFP::SPCS-1 (endoplasmic reticulum, ER in magenta) and mKate::tubulin (in green).

https://elifesciences.org/articles/97812/figures#fig1video1

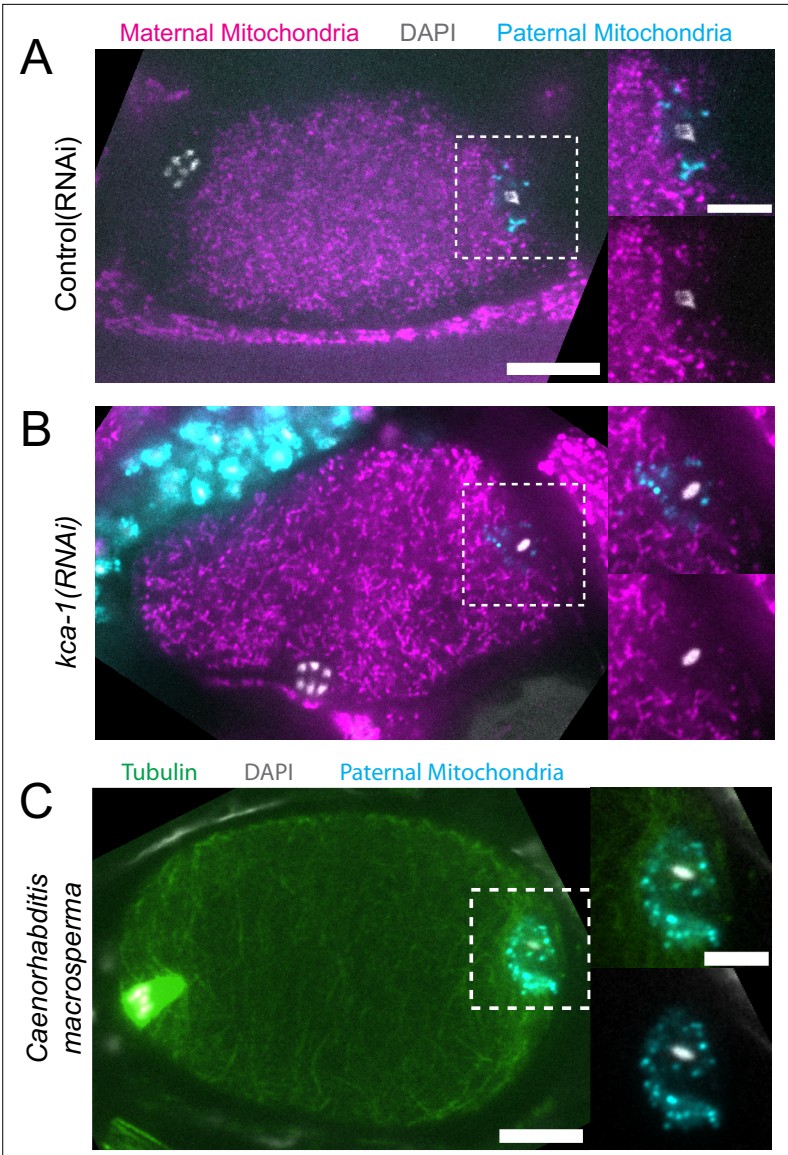

**Figure 2.** Paternal organelle cloud is impermeable to maternal mitochondria regardless of yolk packing and is conserved in Nematoda. (**A**) Fixed image of a meiotic embryo expressing COX-4::GFP (maternal mitochondria) and MitoTracker Deep Red FM (paternal mitochondria). Paternal mitochondria take up a volume excluding maternal mitochondria (n=11). (**B**) Upon depletion of the kinesin cargo adapter, KCA-1, maternal mitochondria extend to the plasma membrane but paternal mitochondria remain sequestered (n=8). (**C**) Fixed image of a *Caenorhabditis macrosperma* (*C. macrosperma* wild isolate) meiotic embryo mated with MitoTracker Deep Red FM stained males. The paternal mitochondria are in a larger volume than in *C. elegans* but the cohesion of mitochondria near paternal DNA is conserved between species (n=4). (**A–C**) Bars: whole embryo 10 μm; inset 5 μm. White dotted boxes denote area of insets.

the meiotic spindle (n=9 metaphase I) were observed in cortical regions that are free of yolk granules (*Figure 1B*). Maternal mitochondria labeled with COX-4::GFP were also packed inward, away from the cortex, and were excluded from the cloud of paternal mitochondria and excluded from the spindle (n=5 metaphase I; *Figure 1C*). Time-lapse imaging (n=21) revealed that the cloud of paternal mitochondria remained together and separate from maternal yolk granules even when moving long distances with cytoplasmic streaming (*Figure 1D*).

Because yolk granules and maternal mitochondria are packed inward during meiosis (*Figure 1B and C*) and sperm must enter from the outside, it is possible that the exclusion of maternal yolk granules from the volume of paternal mitochondria might simply be a consequence of inward packing.

In *kca-1(RNAi)* meiotic embryos, yolk granules (*McNally et al., 2010*) and mitochondria (*Figure 2A and B*) do not pack and instead extend to the plasma membrane. Maternal mitochondria were still excluded from the volume of paternal mitochondria in *kca-1(RNAi)* embryos (n=11 control, 8 RNAi; *Figure 2B*). This result indicated that the volume of paternal mitochondria excludes maternal mitochondria. In addition, paternal mitochondria also formed a discrete cluster in meiotic embryos of a nematode species with giant sperm (*Vielle et al., 2016*) (n=4; *Figure 2C*) indicating that the unique properties of the ball of paternal organelles in the meiotic zygote are conserved.

## Maternal ER invades the volume of paternal organelles shortly after fertilization

In contrast with maternal yolk granules and maternal mitochondria, maternal ER was observed penetrating into the cloud of paternal mitochondria and enveloping the paternal DNA in a shell that appeared as a ring in confocal sections of both live and fixed meiotic embryos (n=10; *Figure 3A*). To elucidate how the maternal ER enters the ball of paternal organelles, we monitored seven instances of sperm-egg fusion by time-lapse microscopy. Previous studies have demonstrated that *C. elegans* sperm fuse with the egg as opposed to being phagocytosed (*Takayama and Onami, 2016*). Mitotracker-labeled males were mated with hermaphrodites with a maternally-expressed ER marker and a maternally-expressed plasma membrane marker (mCherry::PH). 20 s after the apparent entry of paternal mitochondria into the egg (0:20 in *Figure 3B*), a sperm-sized volume devoid of maternal ER was observed within the maternal plasma membrane. One minute after apparent sperm-egg fusion, the maternal ER had moved into the plasma membrane behind the sperm (1:25 in *Figure 3B*; 1:00 in *Figure 3C*). The maternal ER ring enveloping the sperm DNA was not discernible until 7 min after apparent sperm-egg fusion (7:24 in *Figure 3C*), although it might form earlier because the zygote undergoes dramatic movement through two sequential sphincters into the spermatheca then into the uterus during this time. The envelopment of the sperm DNA by maternal ER may be an initial step in nuclear envelope assembly (*Penfield et al., 2020*; *Barger et al., 2023*), which has been halted until after the completion of meiosis. These results show that maternal ER is initially excluded from the sperm at fusion. Since maternal mitochondria and yolk granules are excluded later, this suggests that all maternal membranes are initially excluded from the sperm at fusion. The plasma membrane marker diffuses in behind the sperm first, followed by plasma-membrane-associated ER, followed by envelopment of the sperm DNA by maternal ER by 7 min after fusion. Maternal yolk granules and maternal mitochondria, however, are excluded from the sperm volume for much longer (*Figure 1*).

## The maternal ER envelope around the sperm DNA is permeable to proteins

If the maternal ER envelope around sperm DNA was sealed and impermeable during meiosis, this could both prevent the sperm DNA from inducing ectopic spindle assembly and prevent the sperm DNA from interacting with meiotic spindle microtubules. To test whether the ER envelope around the sperm DNA is permeable to proteins, we analyze BAF-1, a chromatin-binding component of the inner nuclear envelope (*Gotzmann and Foisner, 1999*). Maternally provided GFP::BAF-1 was not detected on maternal or paternal chromatin during metaphase I or metaphase II but labeled the surface of both maternal and paternal chromatin during anaphase I and anaphase II (*Figure 4A, B and C*). Because BAF-1 is a chromatin-binding protein, this result indicates that BAF-1 can pass freely through holes in the ER envelope surrounding the sperm DNA between metaphase I and anaphase I.

## Movement of sperm contents within the zygote is limited by katanin and kinesin-13 and this limitation prevents capture of the sperm DNA by the meiotic spindle

*C. elegans* meiotic cytoplasmic streaming (*McNally et al., 2010*; *Panzica et al., 2017*; *Yang et al., 2003*; *Kimura et al., 2017*) has the potential to bring the sperm contents into close proximity with the meiotic spindle, however, a previous study found that the male pronucleus very rarely forms at the same end of the embryo as the female pronucleus (*Kimura and Kimura, 2020*). This indicates that limitations on cytoplasmic streaming might be involved in maintaining distance between the sperm contents and the meiotic spindle. We, therefore, asked whether increasing cytoplasmic streaming would cause collisions between the meiotic spindle and sperm contents. Meiotic cytoplasmic

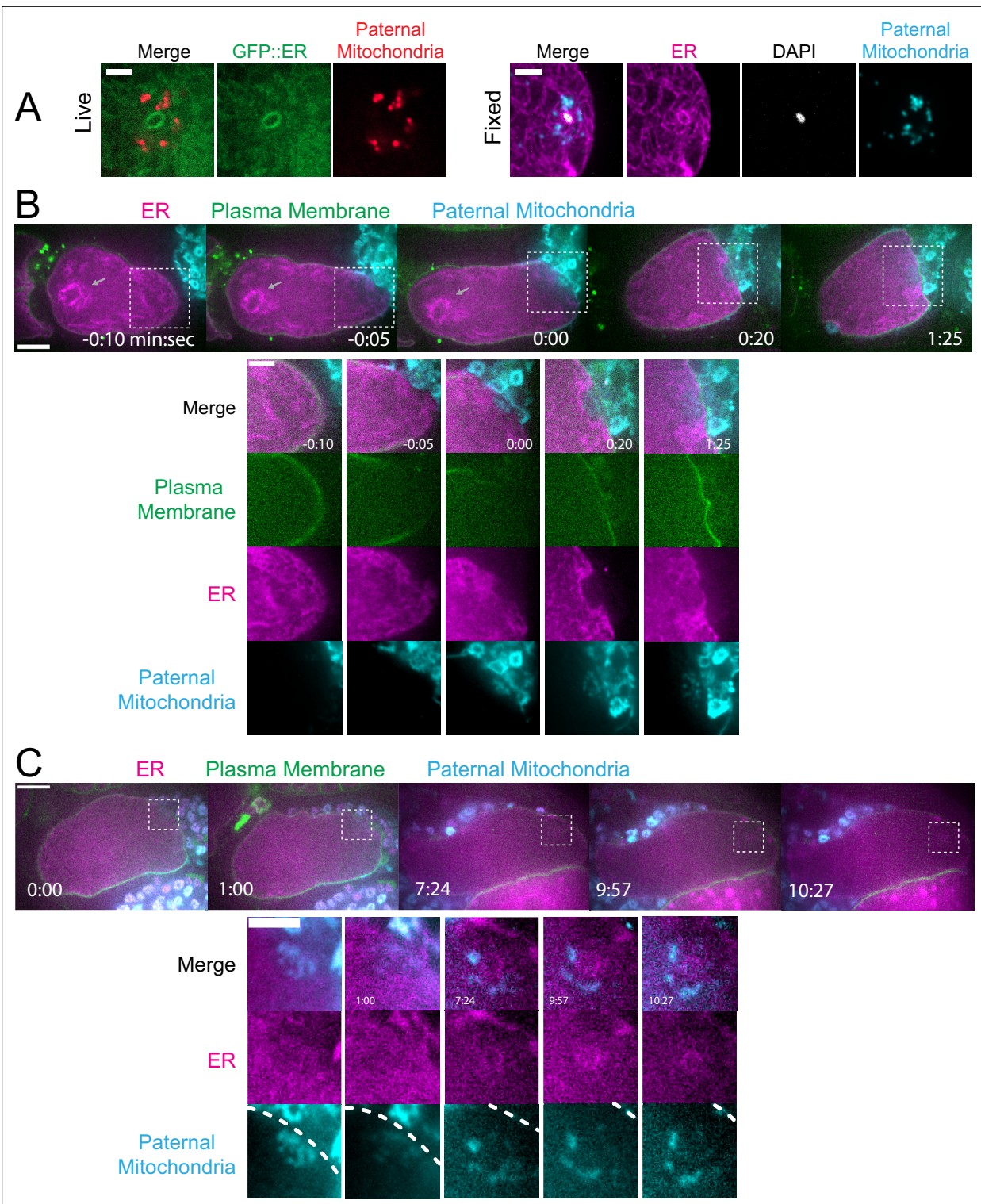

**Figure 3.** Maternal endoplasmic reticulum (ER) enters the sperm cytoplasm after a delay and forms a ring around sperm DNA. (**A**) Live and fixed images of anaphase I and metaphase I embryos show an ER ring within the sperm cytoplasm (as indicated by sperm mitochondria). The fixed image also shows DAPI staining within the ER ring. Live, n=10, fixed n=10. Bars: 3 μm (**B**) Time-lapse images of a strain expressing TMCO-1::GFP (maternal ER) and mCH::PH (maternal plasma membrane) and fertilized by a sperm labeled with MitoTracker Deep Red FM (paternal mitochondria). At fertilization, a gap appears in the mCH::PH as the sperm's membrane fuses with that of the oocyte. Paternal mitochondria are seen inside the oocyte at time 0:00. A 'pocket' in the maternal ER contains the paternal mitochondria (n=7). At time 1:25 the mCH::PH gap has closed and fertilization is complete. Arrows denote prometaphase spindle. (**C**) Time-lapse images of an embryo as it enters the spermatheca and exits into the uterus. ER can be seen forming a

*Figure 3 continued on next page*

*Figure 3 continued*
ring within the mass of paternal mitochondria at 7:24 (n=2). Dotted lines denote cell membrane. White dotted boxes denote area of insets. (**B–C**) Bars: whole embryos 10 µm; insets 5 µm.

streaming requires microtubules (*Yang et al., 2003*) and kinesin-1 (*McNally et al., 2010*). Depletion of katanin by *mei-1(RNAi)* increases the mass of cortical microtubules and the extent of yolk granule streaming (*Kimura et al., 2017*) and depletion of kinesin-13 by *klp-7(RNAi)* also increases the mass of cortical microtubules (*Gigant et al., 2017*). We found that *mei-1(RNAi)* or *klp-7(RNAi)* increased the maximum displacement of the sperm contents in both the long and short axes of the ellipsoid embryo relative to control L4440(RNAi) embryos (*Figure 5A, B, C and D*; *Figure 5—videos 1–3*). The only demonstrably significant difference between the excessive cytoplasmic streaming in these two depletions was that excessive streaming in *klp-7(RNAi)* persists for a longer period of time, well into metaphase II (*Figure 5D*).

Among 15 control L4440(RNAi) time-lapse sequences, the closest center-to-center distance between spindle and sperm contents was 18 µm. In contrast, among 51 time-lapse sequences of *mei-1(RNAi)* meiotic embryos, the sperm came within 5.5 µm center to center distance of the spindle in 12 cases, and in 12/12 of these cases, the sperm stopped moving relative to the spindle indicating

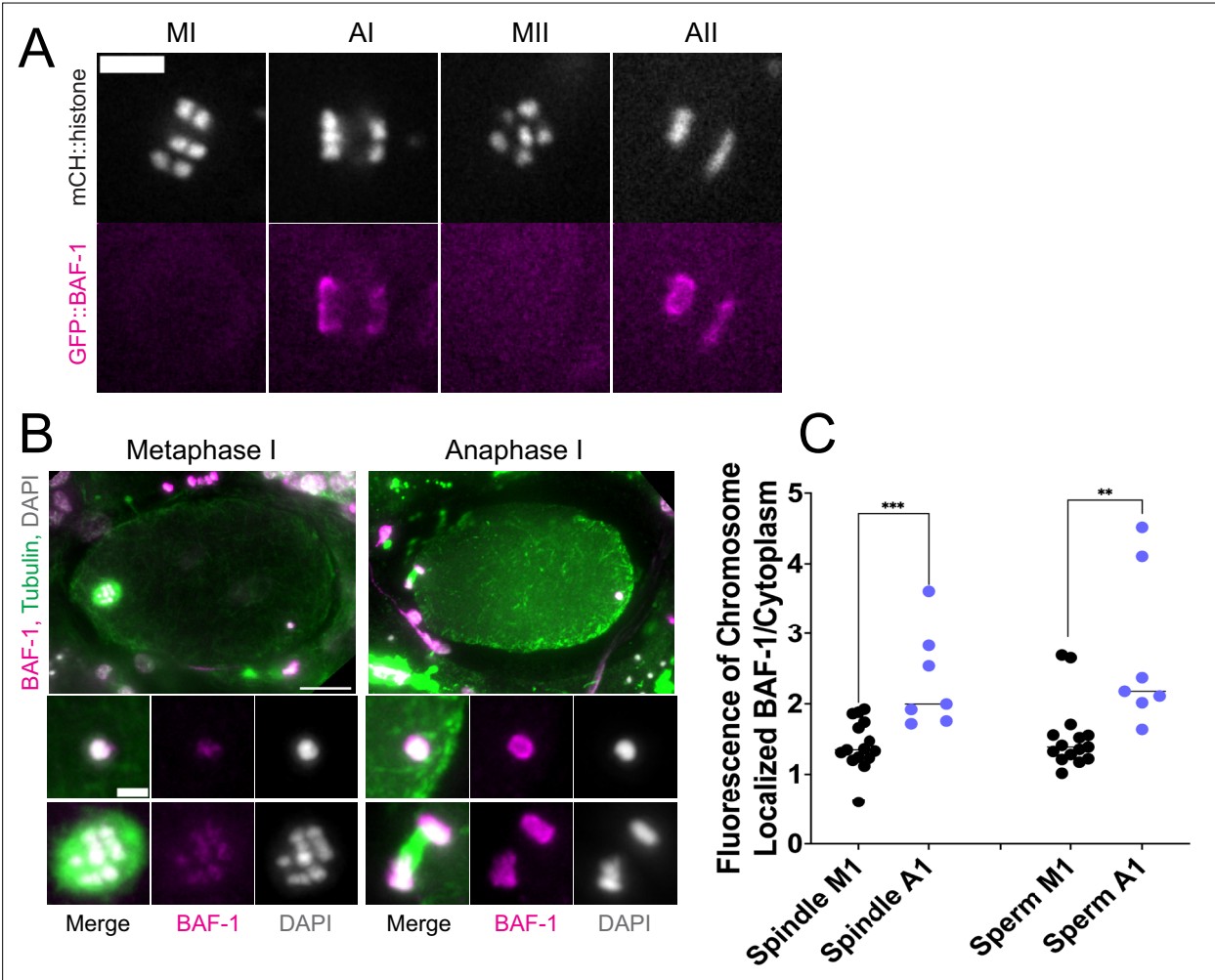

**Figure 4.** Endoplasmic reticulum (ER) surrounding the sperm DNA is permeable to the chromatin-binding protein BAF-1. (**A**) Time-lapse images of maternal chromosomes in an embryo expressing mCH::HIS and GFP::BAF-1. Maternal BAF-1 localizes to chromosomes during anaphase, after the assembly of the ER envelope. Bar, 5 µm. (**B**) In fixed embryos, maternal GFP::BAF-1 strongly localizes to both maternal and paternal chromosomes during anaphase I, but not metaphase I. Bars: whole embryo, 10 µm; inset, 2 µm. (**C**) Ratios of chromosomal to cytoplasmic GFP::BAF-1 show that, during anaphase I, there is an increase in GFP::BAF-1 on both the maternal and paternal chromosomes. **p<0.01, ***p<0.001 by Mann-Whitney U Test.

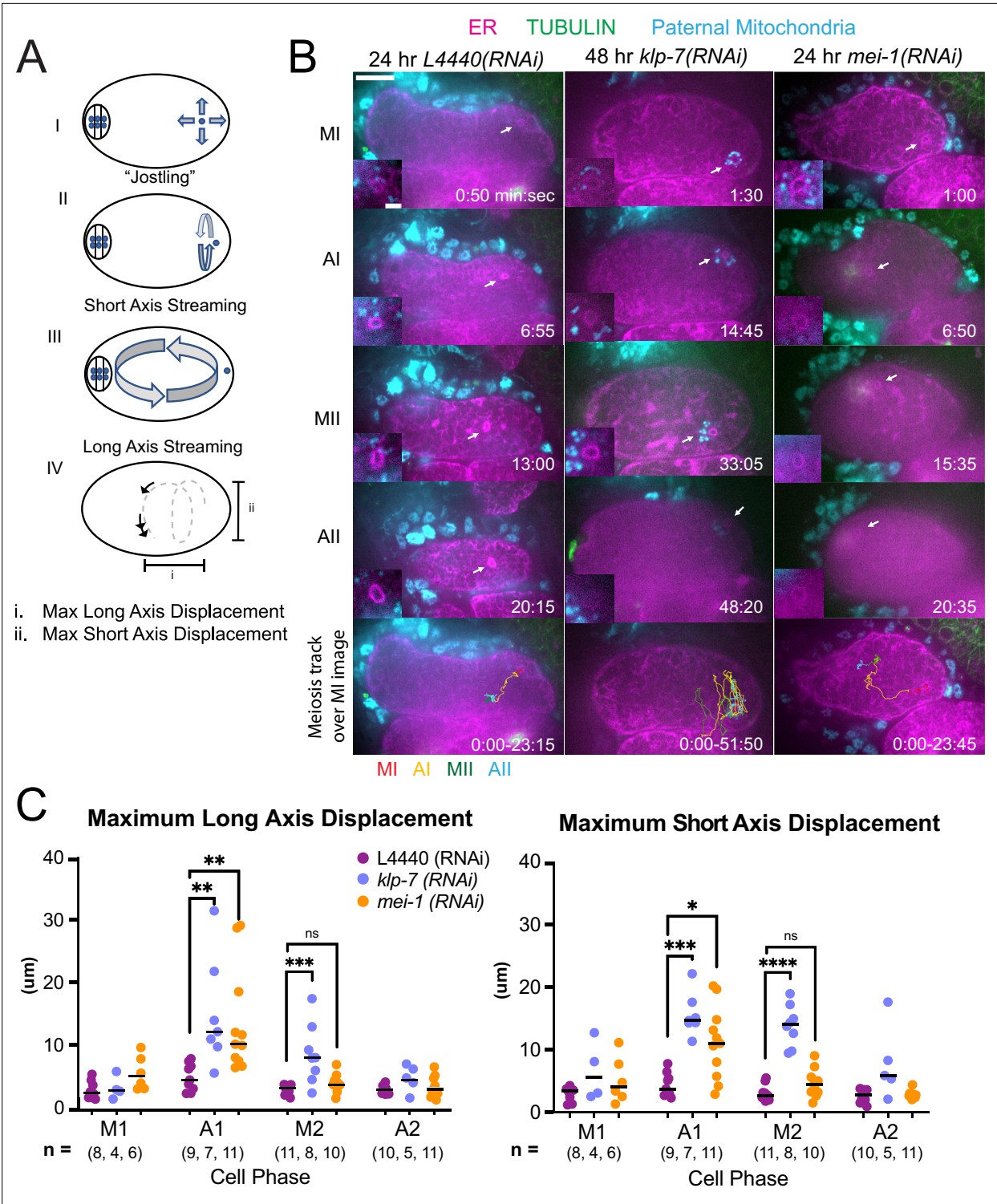

**Figure 5.** MEI-1[katanin] and KLP-7[kinesin-13] limit meiotic cytoplasmic streaming of the sperm contents. (**A**) Illustration of paternal DNA dynamics in meiotic embryo. (I) 'Jostling' denotes random movements without specific direction. (II), (III) Short-axis and long-axis streaming refers to rotation around short and long-axis, respectively. (IV) Illustration of measurements taken from tracks following course of paternal DNA. (**B**) Live imaging of GFP::SP12 (ER); mKATE::TUBULIN embryos with Deep Red MitoTracker stained paternal mitochondria from *fog-2(q71)* males. Timelapse frames after control treatment with L4440(RNAi). Timelapse images after treatment with *klp-7(RNAi)* showing sperm streaming in the short-axis of the embryo. Timelapse images after treatment with *mei-1(RNAi)* showing sperm contents streaming long-axis of the embryo. Tracks show movement of paternal DNA throughout each cell phase. Arrows denote endoplasmic reticulum (ER) ring around paternal DNA. 5 s intervals. Time zero is once embryo fully exited the spermatheca into the uterus. Bars: (whole embryo) 10 μm; (inset) 2 μm. MI, metaphase I; AI, anaphase I; MII, metaphase II; AII, anaphase II. (**C**) Measurements of max x-axis

*Figure 5 continued on next page*

*Figure 5 continued*

and y-axis displacement of tracks following course of paternal DNA throughout different phases. *p<0.05, **p<0.01, ***p<0.001 ****p<0.0001 Kruskal Wallis Test.

The online version of this article includes the following video and figure supplement(s) for figure 5:

**Figure supplement 1.** Distance of the sperm contents from the cortex of control embryos at metaphase I vs anaphase I.

**Figure 5—video 1.** Movement of sperm contents in control L4440(RNAi) meiotic embryo expressing GFP::SPCS-1 (endoplasmic reticulum, ER in magenta), mKate::tubulin (not in focal plane), and paternal mitochondria labeled with Mitotracker Deep Rd (cyan).

https://elifesciences.org/articles/97812/figures#fig5video1

**Figure 5—video 2.** Movement of sperm contents in *mei-1(RNAi)* meiotic embryo expressing GFP::SPCS-1 (endoplasmic reticulum, ER in magenta), mKate::tubulin (green), and paternal mitochondria labeled with Mitotracker Deep Rd (cyan).

https://elifesciences.org/articles/97812/figures#fig5video2

**Figure 5—video 3.** Movement of sperm contents in *klp-7(RNAi)* meiotic embryo expressing GFP::SPCS-1 (endoplasmic reticulum, ER in magenta), mKate::tubulin (green), and paternal mitochondria labeled with Mitotracker Deep Rd (cyan).

https://elifesciences.org/articles/97812/figures#fig5video3

**Figure 5—video 4.** Movement of sperm contents, maternal endoplasmic reticulum (ER), and maternal yolk granules increases at anaphase onset when sheet-like ER disperses.

https://elifesciences.org/articles/97812/figures#fig5video4

a capture event (*Figure 5B*; *Figure 5—video 2*). Among a subset that could be tracked through the pronuclear stage, a single male pronucleus formed adjacent to multiple small female pronuclei. Among 25 time-lapse sequences of *klp-7(RNAi)* embryos, the sperm DNA became stuck against the meiotic spindle (4.6 µm center to center distance) in only one case. In all other cases, the sperm streamed past a stationary spindle. However, the closest sperm to spindle distance was 7.7 µm in one case. These results indicate that limiting cytoplasmic streaming is important for maintaining a distance between spindle and sperm and thus preventing the capture of the sperm by the meiotic spindle. However, *mei-1(RNAi)* spindles are apolar and do not undergo normal polar body extrusion, and the single capture event in a *klp-7(RNAi)* embryo resulted in cell cycle arrest. These results, therefore, do not reveal what would happen to paternal DNA captured by a normal meiotic spindle during polar body extrusion.

An important question is why the sperm contents move more than the meiotic spindle. During metaphase I, when the ER is sheet-like, the movement of the sperm contents is limited. When the ER disperses during anaphase I, the spindle rotates and one pole is moved closer to the cortex by cytoplasmic dynein (*Figure 1—video 1*; *Ellefson and McNally, 2011*). In contrast, the distance of the sperm contents from the cortex (*Figure 5—figure supplement 1*) and the movement of the sperm contents both increase when the ER disperses (*Figure 5C*; *Figure 5—video 4*).

## Ataxin-2 is required to maintain the cohesion of paternal mitochondria

Because maternal ER penetrates the sperm contents and envelops the sperm DNA, and because the *C. elegans* ortholog of ataxin-2, ATX-2, affects ER organization (*Del Castillo et al., 2019*), we analyzed the contribution of ATX-2 to the cohesion of the sperm contents in meiotic embryos. We first introduced an auxin-induced degron and GFP tag to the 3'end of the endogenous *atx-2* gene. Endogenously tagged ATX-2 was observed throughout oocytes and meiotic embryos (*Figure 6B and C*; *Figure 6—figure supplement 1*). ATX-2 did not uniquely co-localize with ER (*Figure 6—figure supplement 1*). Dark holes were observed suggesting exclusion from the lumens of larger membranous organelles (*Figure 6C*; *Figure 6—figure supplement 1*). We then compared the intensity of the GFP signal after three different depletion treatments, 1- hr auxin, 24 hr *GFP(RNAi)*, or 27 hr *atx-2(RNAi)*. All three methods resulted in a significant reduction of the GFP signal (*Figure 6B, C and D*).

The extent of scattering of paternal mitochondria was analyzed in Z-stacks of fixed meiotic embryos depleted of ATX-2 by each of the three methods. This analysis was restricted to embryos from anaphase I through anaphase II because our streaming data (*Figure 5C*) and that of Kimura (*Kimura and Kimura, 2020*) indicate that the sperm contents had not moved significantly before anaphase I. Example Z-projections are shown in *Figure 7*. Mitochondria can exist as a tubular network (*Okamoto and Shaw, 2005*) and tubules in close proximity cannot always be resolved by light microscopy. To account for the apparent heterogeneous sizes of foci of paternal mitochondria, the

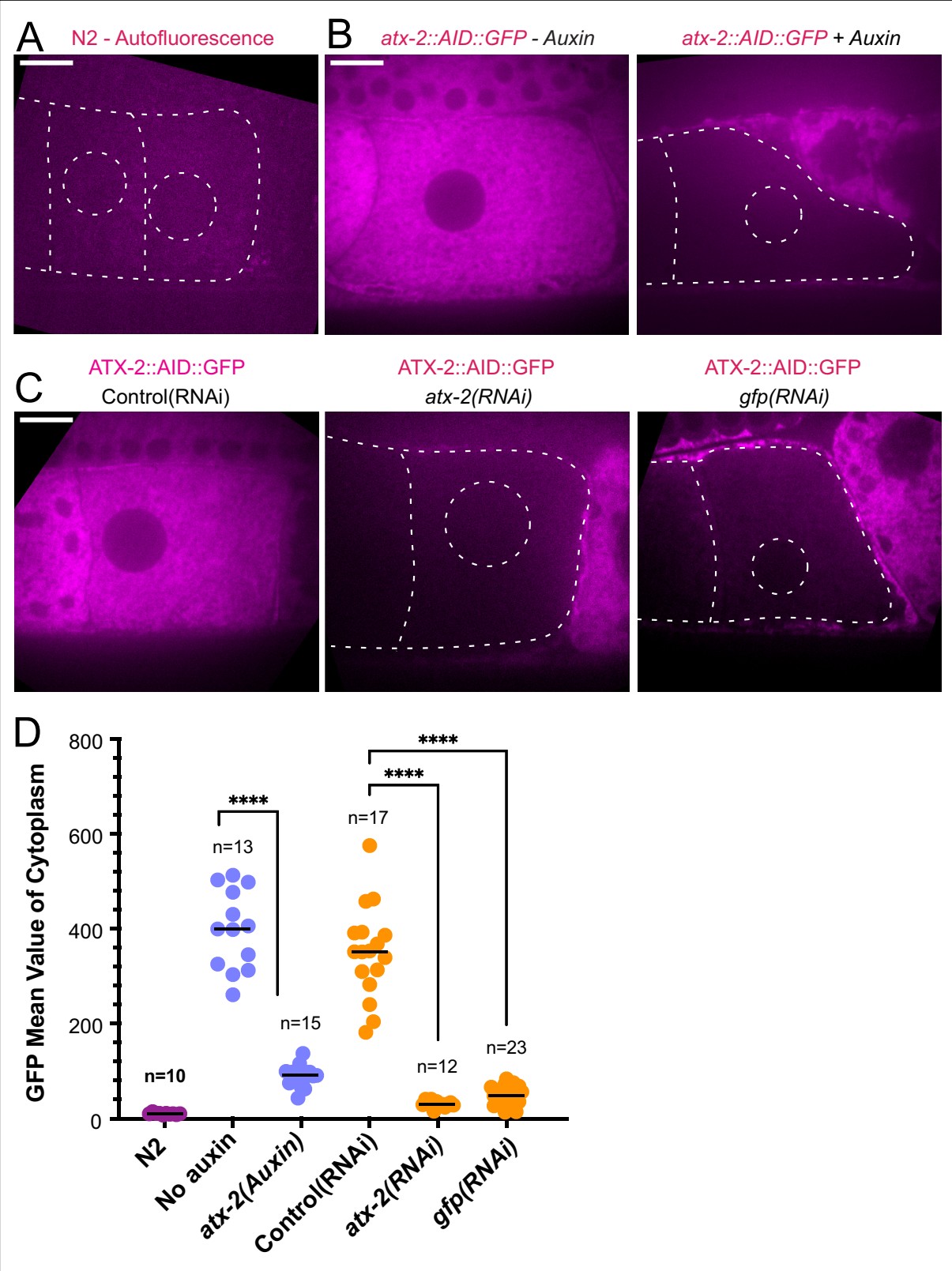

**Figure 6.** ATX-2 is depleted by three different methods. (**A**) Single plane live image of N2 (no GFP). Autofluorescence shows dim outline of oocytes and germinal vesicle. (**B–C**) Single plane live images of –1 oocytes in strain with ATX-2::AID::GFP. No auxin treatment shows endogenous GFP-tagged ATX-2 fluorescence throughout the cytoplasm. Auxin treatment results in depletion of ATX-2 in oocytes. (**C**) Live images of –1 oocytes in strain with ATX-

*Figure 6 continued on next page*

*Figure 6 continued*

2::AID::GFP. Control L4440(RNAi) shows ATX-2 throughout the cytoplasm, but ATX-2 is depleted after 27 hr *atx-2(RNAi)* or 24 hr *gfp(RNAi)*. (**A–C**) Bars, 10 μm. (**D**) Mean GFP fluorescence in the cytoplasm of –1 oocytes after each treatment. ****p<0.0001 by Welch's t-test and Brown-Forsythe test.

The online version of this article includes the following figure supplement(s) for figure 6:

**Figure supplement 1.** Localization of ATX-2 in –1 oocyte and +1 meiotic embryo.

fluorescence intensity of larger foci was divided by the fluorescence intensity of the smallest foci such that a focus with three times the fluorescence intensity was counted as three 'mitochondria.' ATX-2 depletion by each of the three methods resulted in a significantly increased mean distance of paternal mitochondria from the sperm DNA (*Figure 8A*) as well as significantly increased standard deviation of individual distances (*Figure 8B*). Standard deviation is an important measure because an increase in the number of mitochondria both closer and further from the sperm DNA would result in no change in the mean. It remains possible that paternal mitochondria scatter is caused by pressure applied during fixation, pressure resulting from ovulation through the spermatheca valves, or cytoplasmic streaming. We were not able to unambiguously track scattering by live imaging because significant movement occurs during the acquisition of a complete z stack and because the embryos cannot withstand the additional photodamage. These results still support the hypothesis that ATX-2 is required to maintain the integrity or cohesiveness of the ball of paternal mitochondria that surrounds the sperm DNA to resist external forces.

## Double depletion of KLP-7 and ATX-2 results in the capture of the sperm DNA by the spindle

Because we observed increased cytoplasmic streaming in *klp-7(RNAi)* and disruption of the integrity of the sperm contents after ATX-2 depletion, we hypothesized that double depletion of KLP-7 and ATX-2 would result in an increased frequency of spindle microtubules capturing the sperm DNA. Among 24 time-lapse sequences of *atx-2(AID +auxin) klp-7(RNAi)* meiotic embryos, the ER envelope around the sperm DNA moved extensively in all cases but came within a threshold distance of 5.5 μm (center to center) of the meiotic spindle in only five cases. In 5/5 of these cases, a bundle of microtubules extended from the meiotic spindle into the ER envelope around the sperm DNA, and the sperm DNA became stuck in this position (*Figure 9A and B*). In one case, the ER envelope around the sperm DNA was transiently captured by the meiosis I spindle and stretched in the direction of streaming (*Figure 9C*). The sperm DNA then released, continued streaming, and was captured by the meiosis II spindle. All cases of stable capture caused a cell-cycle arrest so that the consequences of polar body extrusion could not be determined.

Because ATX-2 depletion alters ER morphology, we were not able to score cytoplasmic streaming with the cell-cycle accuracy shown in *Figure 5*. However, the maximum long-axis and short-axis displacement at any cell-cycle time was increased in *atx-2(AID)* with auxin vs without auxin (*Figure 9—figure supplement 1*). No spindle capture events were observed, however, among 15 ATX-2 single depletion time-lapse sequences and the closest distance between sperm and spindle was 8.2 μm with 1 hr auxin and 19.3 μm without auxin.

The frequency of sperm capture by the meiotic spindle (*Figure 9D*) was significantly higher than wild-type controls in *klp-7(RNAi) atx-2(AID)* double-depleted embryos (p=0.011 Fisher's exact test). Although the number of single mutant embryos analyzed was too low to demonstrate a significant difference between single and double mutant embryos, these results qualitatively support the hypothesis that limiting cytoplasmic streaming and maintaining the integrity of the ball of paternal mitochondria are both important for preventing capture events between the meiotic spindle and sperm DNA.

## Discussion

The mechanism excluding maternal yolk granules and mitochondria from the volume of sperm cytoplasm introduced to the egg at fertilization is not clear. The simplest hypothesis is that maternal and paternal cytoplasm might not mix during the 45 min from GVBD to pronucleus formation due to the high viscosity of cytoplasm. Attempts at measuring the cytoplasmic viscosity of the *C. elegans* zygote have revealed values from 0.67 to 1.0 Pa s (*Khatri et al., 2022*; *Daniels et al., 2006*; *Garzon-Coral*

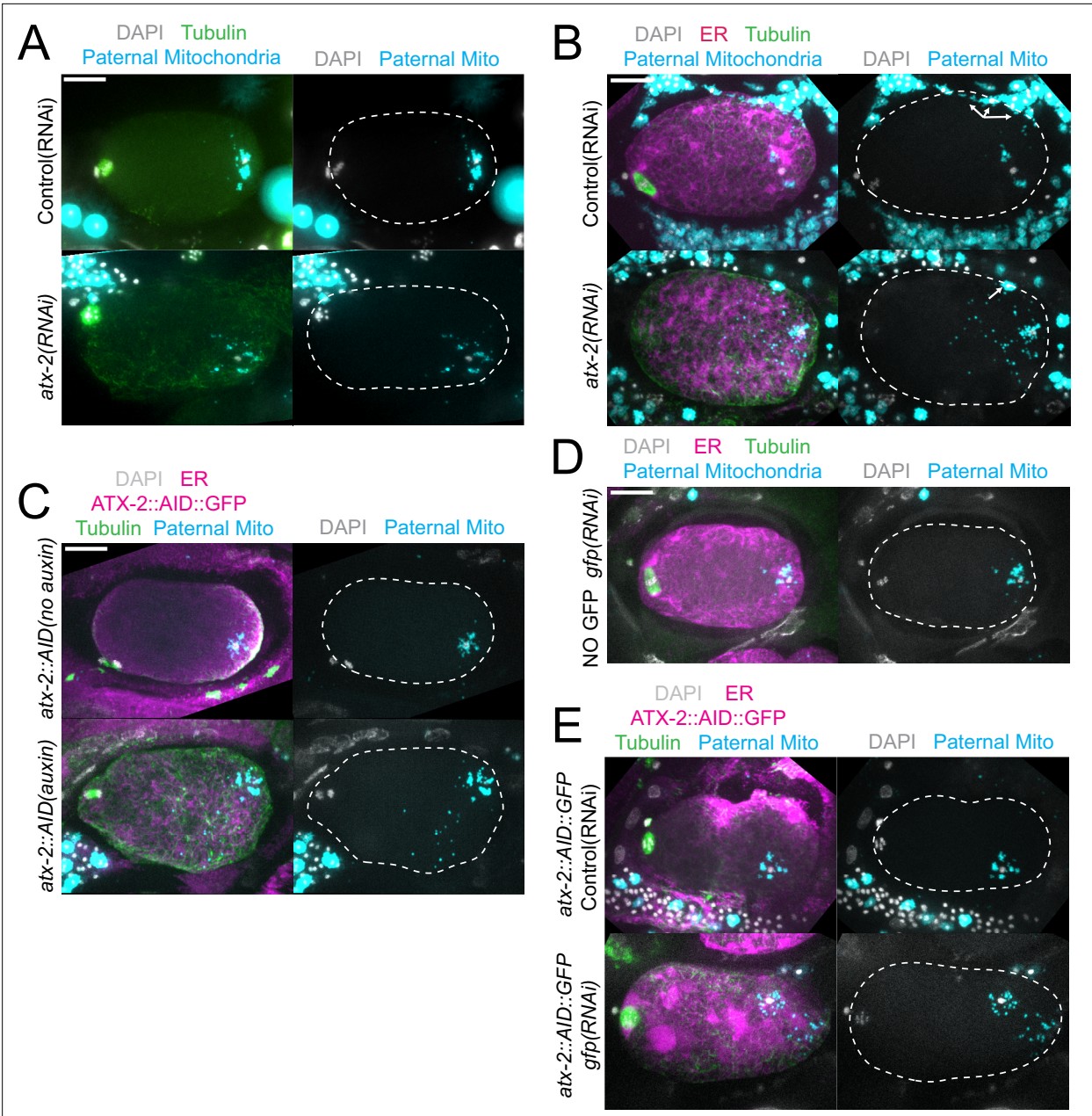

**Figure 7.** Paternal mitochondria scatter during meiosis after ATX-2 depletion. (**A–E**) Maximum intensity projections of z-stacks of fixed meiotic embryos stained with tubulin antibodies and DAPI and with paternal mitochondria labeled by mating with MitoTracker Deep Red FM treated *fog-2(q71)* males. (**A**) N2 wild-type embryos treated with control L4440(RNAi) or *atx-2(RNAi)*. (**B, C, E**) Embryos expressing TIR1, GFP::SPCS-1/ER, mKate::TBA-2, and with endogenously tagged ATX-2::AID::GFP. (**B**) 27 hr control L4440(RNAi) or *atx-2(RNAi)*. Arrows denote mitochondrial fluorescence from sperm outside the embryo overlapping with the embryo as a result of the maximum intensity projection. (**C**) No auxin or 1 hr auxin treatment. (**D**) GFP(RNAi) on strain with no tag on ATX-2 but expressing GFP::SPCS-1/ER and mKate::TBA-2. GFP::SPCS-1/ER fluorescence remains because SPCS-1 and ATX-2 are tagged with GFPs with different sequences. (**E**) Control L4440(RNAi) or gfp(RNAi) of ATX-2::AID::GFP strain. Bars, 10 μm. White dotted outlines indicate the cortex of the cell.

*et al., 2016*) which are similar to the viscosity of 100% glycerol. Alternatively, the sperm contents might be held together by a cytoskeleton-like matrix as proposed for the Balbiani body (*Boke et al., 2016*). In either case, an active process appears to allow the maternal ER to penetrate into the paternal cytoplasm to envelope the sperm DNA. The capture of the sperm DNA by the meiotic spindle in ATX-2 KLP-7 double-depleted embryos (*Figure 9*) suggests that the integrity of the exclusion zone around the sperm DNA might insulate the sperm DNA from spindle microtubules. However, a much larger

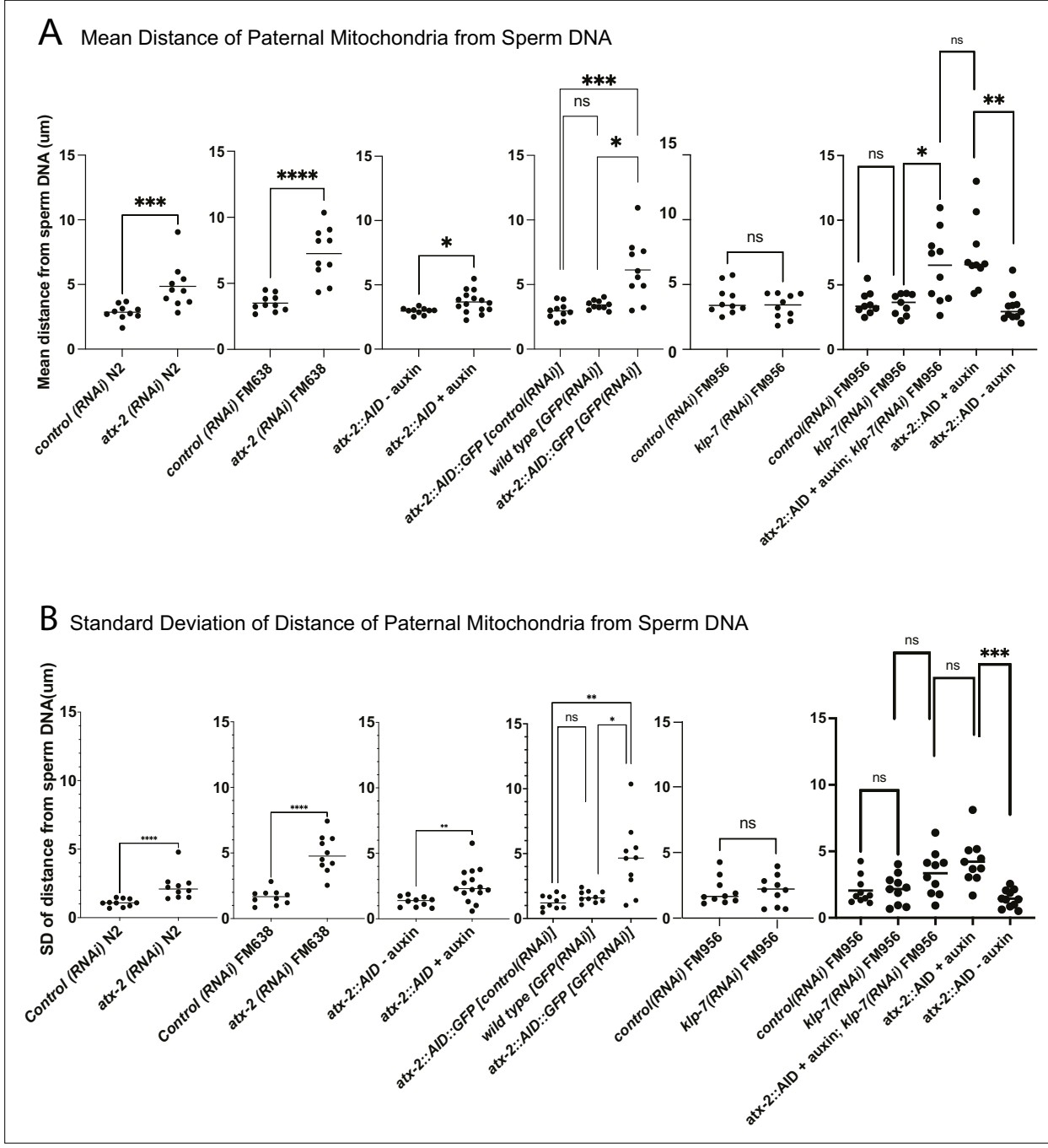

**Figure 8.** Quantification of paternal mitochondrial scatter in ATX-2-depleted anaphase I meiotic embryos. (**A**) Mean and (**B**) standard deviation of the distance of individual paternal mitochondria from the sperm DNA determined from Z-stacks of fixed anaphase I embryos. Each dot represents one embryo. Distances for individual mitochondria are in Supplementary data file.

number of *klp-7(RNAi)* singly depleted and *atx-2(degron)* singly depleted time-lapse sequences are needed to rigorously support this idea.

ATX-2 is required to maintain the integrity of the ball of paternal mitochondria around the sperm DNA (*Figures 7 and 8*), but the mechanism is unknown. Because the paternal mitochondria observed to scatter are from wild-type males, the effect of ATX-2 depletion must be on the egg and not on the sperm. Although ATX-2 depletion alters ER morphology (*Del Castillo et al., 2019*) we still observed a maternal ER envelope around the sperm DNA in all ATX-2-depleted embryos. ATX-2 also plays roles in translational regulation (*Ciosk et al., 2004*), germline proliferation (*Maine et al., 2004*), cytokinesis

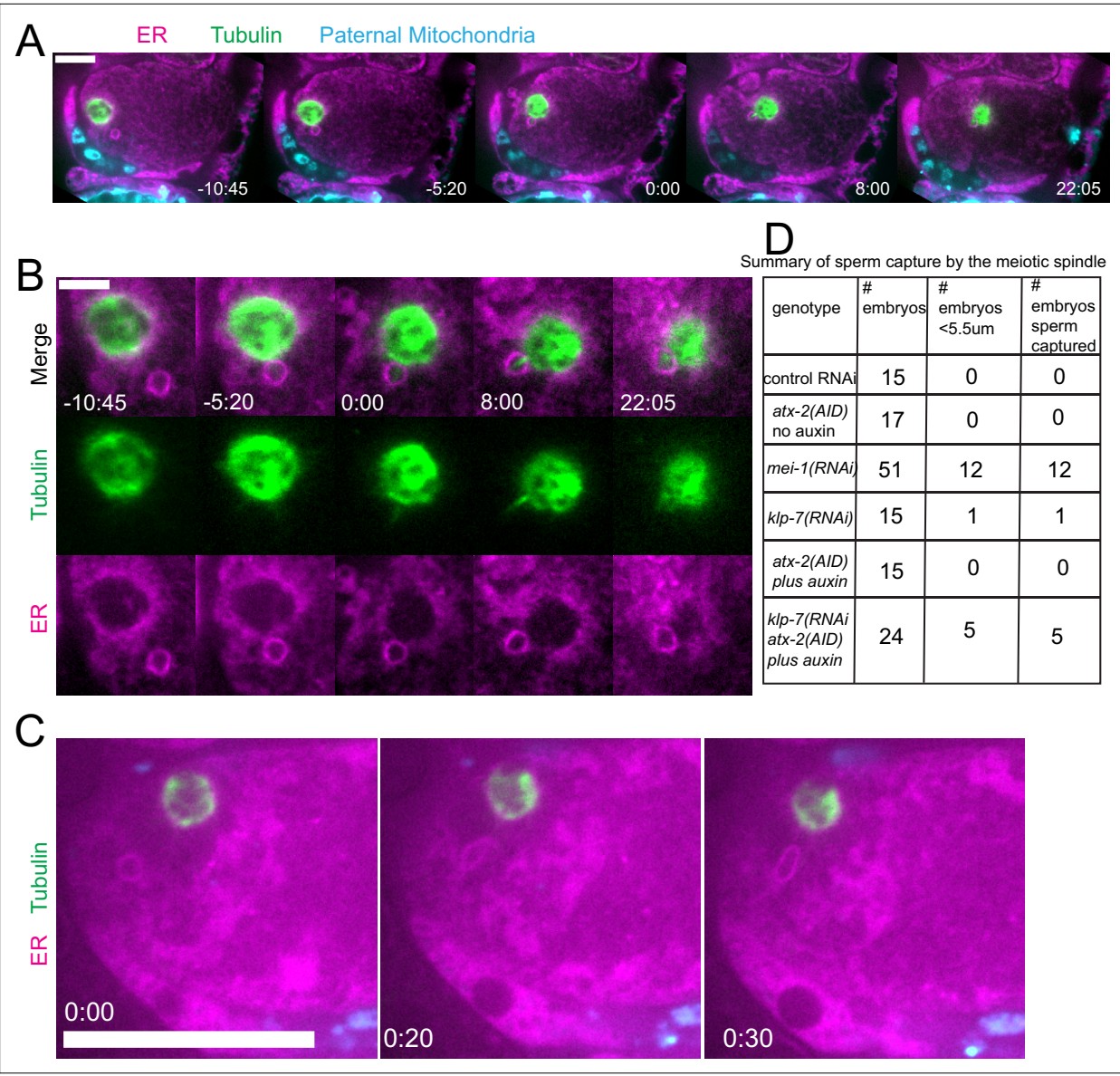

**Figure 9.** Capture of the sperm DNA by the meiotic spindle in KLP-7 ATX-2 double depleted meiotic embryos. (**A**) Live imaging of GFP::SP12/ ER; mKATE::TUB; ATX-2::AID::GFP meiotic embryo with Deep Red MitoTracker stained paternal mitochondria from *fog-2(q71)* males mated in. Hermaphrodites were treated with *atx-2/klp-7 (auxin/RNAi)*. 5/24 videos had sperm ring travel within <5.5 μm of spindle (measured center to center). Of those videos, 5/5 resulted in a microtubule bridge between the spindle and the middle of the sperm ring. (**B**) Higher magnification of (**A**). (**C**) Example of endoplasmic reticulum (ER) ring around the sperm DNA stretching toward the spindle. (**D**) Summary of capture events. Bars: (whole embryo) 10 μm; (inset) 5 μm.

The online version of this article includes the following figure supplement(s) for figure 9:

**Figure supplement 1.** Cytoplasmic streaming after ATX-2 depletion.

(*Gnazzo et al., 2016*), centrosome size (*Stubenvoll et al., 2016*), and fat metabolism (*Bar et al., 2016*). Thus the effects of ATX-2 could be extremely indirect. Because *C. elegans* ovulate every 23 min (*McCarter et al., 1999*), however, our rapid 1 hr depletion would only affect the three most mature oocytes. A speculative possibility is that ATX-2 regulates 'the integrity of the cytoplasm' analogous to the action of ANC-1 in the *C. elegans* hypodermis (*Hao et al., 2021*). Because ATX-2 is an RNA-binding protein with intrinsically disordered domains, it might act as part of the RNA-dependent ER-associated TIS granule network (*Ma et al., 2021*; *Ma and Mayr, 2018*).

In control embryos, the sperm contents rarely came near the meiotic spindle (18 μm closest distance, see text above) in agreement with a previous study that found that male and female pronuclei rarely form next to each other (*Kimura and Kimura, 2020*). Streaming of the sperm contents was most commonly restricted to a jostling motion with little net displacement, circular streaming in the short-axis of the embryo, or long-axis streaming in which the sperm turned away from the spindle before the halfway point of the embryo (*Figure 5*). Depletion of MEI-1 or KLP-7 resulted in longer excursions of the sperm contents in the long-axis of the embryo toward the spindle (*Figure 5*) but frequent capture of the sperm by the spindle was only observed in *mei-1(RNAi)* (*Figures 5 and 9D*). This may be because *mei-1(RNAi)* affects the positioning of the spindle within the embryo (*Yang et al., 2003*) or because the altered structure of the *mei-1(RNAi)* spindle allows spindle microtubules to capture chromosomes further from the spindle center. In capture events observed after double depletion of ATX-2 and KLP-7, a bundle of microtubules was discernible extending from the spindle into the ER envelope surrounding the sperm DNA (*Figure 9*). Such bundles were not observed in *mei-1(RNAi)* capture events (*Figure 5*), likely because of the previously reported low density of microtubules in *mei-1(RNAi)* spindles (*McNally et al., 2006*; *Srayko et al., 2006*).

To our knowledge, the only reported example of premature interaction between the sperm contents and the oocyte meiotic spindle outside of *C. elegans* is the extrusion of sperm DNA into the second polar body when sperm was injected adjacent to the mouse metaphase II spindle (*Mori et al., 2021*). Close proximity of the sperm contents to the meiotic spindle and apparent capture events in *C. elegans* has been reported in only 20–25% of *mei-1(RNAi)*, *klp-7 atx-2* double depletion (*Figure 9D*) or *kca-1(RNAi)* (*McNally et al., 2012*) meiotic embryos. In the case of *kca-1(RNAi)*, premature sperm asters were implicated in the capture events (*McNally et al., 2012*), however, *kca-1(RNAi)* blocks cytoplasmic streaming (*McNally et al., 2010*) and thus likely reduces the probability of the sperm contents moving close to the spindle, which is also mispositioned in *kca-1(RNAi)* (*Yang et al., 2005*). Close proximity of the sperm DNA with the meiotic spindle was also reported in a small fraction of fixed *pfn-1(RNAi)* embryos where increased mobility of the sperm contents was also reported but actual capture events were not reported (*Panzica et al., 2017*). It should be noted that the large volume of oocytes may contribute to the rarity of sperm/spindle capture events. An 80- μm diameter mouse zygote has a volume 16 times greater than a 50-μm × 25- μm ellipsoid *C. elegans* zygote and a 120 μm diameter human zygote has a volume 55 times greater than a *C. elegans* zygote [using the equation $4/3 \, \Pi (r^a)(r^b)(r^c)$].

None of our time-lapse sequences of sperm capture by the meiotic spindle resulted in the extrusion of paternal DNA into a polar body as was observed when sperm was injected next to the meiotic spindle of mouse oocytes (*Mori et al., 2021*), likely because of the pleiotropic effects of depleting MEI-1, KLP-7, or ATX-2. Many of the reported *kca-1(RNAi)* capture events also resulted in an arrest (*McNally et al., 2012*). Future development of more specific perturbations of cytoplasmic streaming and the organelle exclusion zone around the sperm DNA should address this problem.

# Materials and methods

## *C. elegans* strains

Genotypes of strains used in this study are listed in *Supplementary file 1*. *atx-2(syb5389; ATX-2::AID::GFP)* was generated by SunyBiotech using CRISPR/Cas9.

## Live-in-utero imaging

Adult hermaphrodites were anaesthetized with tricaine/tetramisole in PBS as described (*McCarter et al., 1999*; *Kirby et al., 1990*) and then placed on 2% agarose pads on slides. Extra anesthetic was gently pipetted onto the agarose pad and a coverslip was placed on top. The slide was inverted and placed on the stage of an inverted microscope. Meiotic embryos were identified by bright-field microscopy before initiating time-lapse fluorescence. For all live imaging, the stage and immersion oil temperature were 21°C–24°C. For all time-lapse data, single-focal plane images were acquired with a Solamere spinning disk confocal microscope equipped with an Olympus IX-70 stand, Yokogawa CSU10, either Hamamatsu ORCA FLASH 4.0 CMOS (complementary metal oxide semiconductor) detector or Hamamatsu ORCA-Quest qCMOS (quantitative complementary metal oxide semiconductor) detector, Olympus 100 x UPlanApo1.35 oil objective, 100 mW Coherent Obis lasers (405,

640, 488, 561 nm) set at 30% power, and MicroManager software control. Pixel size was 65 nm for the ORCA FLASH 4.0 CMOS detector and 46 nm for the ORCA-Quest qCMOS detector. Exposures were 200ms for the ORCA FLASH 4.0 qCMOS detector and 100ms for the ORCA-Quest qCMOS detector. Time interval between image pairs or trios was 5 s. Focus was adjusted manually during time-lapse imaging.

For the ATX-2 images in *Figure 6*, z-stacks of –1 oocytes of anesthetized live worms were captured with a Zeiss LSM 980 confocal microscope with Airyscan 2 and a Zeiss Objective LD LCI Plan-Apochromat 40 x/1.2 Imm Corr DIC M27 for water, silicon oil or glycerine.

## Fixed immunofluorescence

*C. elegans* meiotic embryos were extruded from hermaphrodites in 0.8x egg buffer by gently compressing worms between a coverslip and a slide, flash frozen in liquid N2, permeabilized by removing the coverslip, and then fixed in ice-cold methanol before staining with antibodies and DAPI. The primary antibodies used in this work were mouse monoclonal anti-tubulin (DM1α; Thermo Fisher Scientific; 1:200) and rabbit anti-GFP (NB600-308; Novus Biologicals; 1:600). The secondary antibodies used were Alexa Fluor 488 anti-rabbit (A-21206; Thermo Fisher Scientific; 1:200), Alexa Fluor 488 anti-mouse (A-21202; Thermo Fisher Scientific; 1:200), and Alexa Fluor 594 anti-mouse (A-21203; Thermo Fisher Scientific; 1:200). Z-stacks were captured at 1 μm steps for meiotic embryos using the same microscope described above for live imaging.

## Sperm ER ring filming

For sperm ER ring filming, the ring was observed throughout meiosis I and II and manually kept in the focal plane by adjusting the stage (z-axis). Filming typically began at spermatheca exit or meiosis I and ended at pronuclear formation or cell arrest. Due to the rigors of filming an embryo with optimal orientation and positioning in the uterus, videos that started and ended mid-phases were still filmed and included in measurements so long as a complete phase was included between starting and ending filming (e.g. AI and MII were measured in videos starting mid-MI and ending mid-AII). In these cases when filming started mid-phase and/or ended prematurely mid-phase, the incompletely filmed phases were not included in quantifications. The phases at which cell arrest occurred were not included in quantifications, but the phases prior were still used.

## Sperm ER ring tracking

Measurements of the ER rings' dynamics in videos of embryos were tracked manually using the Fiji plugin MTrackJ. All embryos quantified were rotated such that at the beginning of each video the pole containing the meiotic spindle was at the left. Only embryos measured at >35 μm long were used for ER ring tracking. Embryos measured <35 μm were tilted and not ideal for tracking. The starting and ending frames of each cell phase were then determined based on ER morphology. Individual tracks for each phase were manually made by clicking on the center of the ER ring, or in cases when it was briefly out of focus, the mitochondria that most closely followed where the ER ring was previously seen. The following criteria was used for determining phases to be quantified by tracks:

> Metaphase I: Starts when the embryo becomes stationary after exiting spermatheca into the uterus and ends when ER reticulation becomes dispersed.
> Anaphase I: Starts when ER becomes dispersed and ends when it begins to reticulate.
> Metaphase II: Starts when ER begins to reticulate and ends when it becomes dispersed.
> Anaphase II: Starts when ER becomes dispersed and ends at paternal pronuclear formation.
> Maximum X-axis displacement was quantified by subtracting the minimum x-axis coordinate of a phase's track from the maximum x-axis coordinate.
> Maximum Y-axis displacement was quantified by subtracting the minimum y-axis coordinate of a phase's track from the maximum y-axis coordinate.

Displacement was measured as the distance between the first point of a phase's track and the last point.

Distance Traveled was measured as the total length of the track that the sperm ring traveled.

Maximum Average Velocity was measured by taking the maximum of 3-point moving averages of velocities in each phase. MTrackJ was used to measure the velocity of the ER ring between each video frame.

Duration of Phases were calculated by subtracting the first frame number of a phase from the last one and multiplying by 5 due to the 5- s intervals.

## Paternal mitochondria scattering quantifications

Paternal mitochondrial scattering was calculated by using a circular ROI to measure the mean value fluorescence of Deep Red MitoTracker FM or Red MitoTracker CMXros labeling paternal mitochondria in meiotic embryos. Mean pixel values were taken using a circle ROI with an 18-pixel diameter to cover the entire area of a region of fluorescence that appeared as a punctum (unless specified otherwise due to a different resolution). This was repeated until all puncta fluorescence in an embryo were measured. For every embryo, a region of cytoplasm without mitochondria had its fluorescence measured. This value was subtracted from every mitochondrial measurement to correct for noise. The distances between the center of each punctum and the center of the paternal DNA were then recorded with a line tool. The Pythagorean theorem was used to determine the distance when the mitochondria were in a different z-stack than the paternal DNA. These distances were recorded in column scatter graphs to observe the distribution of mitochondrial distance from sperm DNA quantitatively. To measure large amorphous masses of mitochondria, the same sized ROI measuring fluorescence in distinct puncta was used to cover a portion of the mass, make measurements, and then moved to another portion to make more measurements. This was repeated until the entire area of the mass had its fluorescence measured. The corresponding distances to the paternal DNA were recorded for every ROI used. The mean value fluorescence of the amorphous masses were typically much greater than the distinct puncta and as such presumably had a greater density of mitochondria per ROI than the puncta. In order to account for this density, the mean values measured from the ROI's over the masses were divided by the average of the mean values of all of the puncta in an embryo. This resulting number was then used to determine how many times the distance between the paternal DNA and the ROI was recorded in the column scatter graphs (e.g. if the rounded value was 2 then the distance was recorded twice). The averages and standard deviations of each distribution of mitochondrial distances from paternal DNA in the embryos were then measured in order to compare the scattering of mitochondria in different experimental treatments. Quantifications were done in meiotic embryos of all phases except metaphase I. Since this phase is right after fertilization, we believe embryos do not have the time to exhibit scattering as a phenotype.

## Paternal mitochondria labeling

L4 *fog-2(q71)* males were picked onto an OP50 plate and treated with 200 μL 0.05 mM working stock of MitoTracker Deep Red FM (Invitrogen) or MitoTracker Red CMXros (Invitrogen) in M9 overnight. Subsequently, all the males were moved to a fresh OP50 plate and incubated for 15 minu to 'wash' away excess MitoTracker. This washing step was conducted three times to fully remove excess MitoTracker. After the third wash, the males were then moved to plates with hermaphrodites 24 hr before they were to be filmed or used for immunofluorescence. A minimum ratio of 1:1 males to hermaphrodites was used for matings.

L4 *sdhc-1::mCherry; him-5(e1490)* males were added to plates with hermaphrodites 24 hr before they were to be filmed. A minimum ratio of 1:1 males to hermaphrodites was used for matings.

## HALO ligand

At least 20 hr before imaging, hermaphrodites expressing HaloTag were treated with 100 μl of 2.5 μM Janelia Fluor HaloTag Ligand 646 or 549 in M9 added to the bacterial lawn of a 60 mm MYOB agar plate.

## RNA interference

For RNA interference, L4 hermaphrodites were placed on RNAi plates with an RNAi bacterial lawn for a set period of time before being used for live imaging or fixed slides. In 48 hr treatment, the worms were moved to a fresh RNAi plate after 24 hr. RNAi plates were always seeded the day before adding worms. For *mei-1, atx-2, gfp*, and L4440 (*RNAi)*, worms fed on RNAi bacterial lawns for 24–28 hr.

For *klp-7 (RNAi)* and its corresponding control L4440 (*RNAi),* worms fed on RNAi bacterial lawns for 48–52 hr.

## Auxin induced degradation

For auxin-induced degradation, L4's of strains endogenously tagged with auxin-inducible degrons and a TIR1 transgene were placed on a fresh plate of OP50. After 24 hr, the hermaphrodites were moved to auxin plates with lawns of OP50 for 1–3 hr before use in live or fixed experiments. 4 mM auxin plates were made by adding 400 mM auxin (indole acetic acid) in ethanol to molten agar which was then poured and seeded with OP50 bacteria. Depletion of ATX-2::AID::GFP was confirmed by the reduction of ATX-2::AID::GFP signal in –1 oocytes.

## Double RNA interference and auxin-induced degradation

In AID *klp-7 RNAi* experiments, L4's were placed on *klp-7 (RNAi)* and then transferred to a fresh *klp-7 (RNAi)* lawn after 24 hr. After a total of 47 hrs of treatment, the worms were moved to auxin/RNAi plates seeded *with klp-7 (RNAi)*. 1 hr later the worms were then used in live or fixed experiments. Auxin/RNAi plates consisted of 4 mM auxin, 1 mM IPTG, and 200 ug/mL ampicillin in agar. Stocks of 400 mM auxin, 1 M IPTG, and 200 mg/mL ampicillin were added to molten agar and mixed to create the plates.

## Fluorescence intensity measurements

The depletion of atx-2 was measured by taking the mean value of an ROI over the cytoplasm of –1 oocytes, taking caution to not include the nucleus. The z-stack at which the nucleus appeared most in focus was used for measurements in each embryo. The mean value of the cytoplasm was then subtracted by the mean value of an area not containing any part of the worm to correct for noise.

GFP::BAF-1 fluorescence was measured by manually tracing the outline of chromosomes during metaphase I and II and then taking the mean value fluorescence. The same ROI was also used to measure the mean value fluorescence of the middle of the embryo's cytoplasm. The mean value of the fluorescence over the chromosomes was then divided by the cytoplasmic mean value in order to measure and compare BAF-1 fluorescence in metaphase I vs anaphase I.

## Statistics

Shapiro-Wilks tests through GraphPad Prism were used to test for normality in all data in which statistical tests were used to compare means. If the test determined the data was normal, p-values were calculated in GraphPad Prism using Welch's T-tests for comparing means of only two groups and ANOVA tests for comparing means of three or more groups. If the data was not normal, p-values were calculated in GraphPad Prism using Mann-Whitney tests for comparing means of only two groups and Kruskal-Wallis tests for comparing means of three or more groups.

## Acknowledgements

This work was supported by the National Institute of General Medical Science grant R35GM136241 to FJM. We thank the *Caenorhabditis* Genetics Center, which is funded by the NIH Office of Research Infrastructure Programs (P40 OD010440), for strains. We thank Marie Kim and Aastha Lele for assistance with mitochondrial scatter quantification.

## Additional information

### Funding

| Funder | Grant reference number | Author |
| --- | --- | --- |
| National Institute of General Medical Sciences | R35GM136241 | Francis J McNally |
| United States Department of Agriculture/National Institute of Food and Agriculture Hatch Project | 1009162 | Francis J McNally |

| Funder | Grant reference number | Author |
|---|---|---|

The funders had no role in study design, data collection and interpretation, or the decision to submit the work for publication.

## Author contributions
Elizabeth A Beath, Data curation, Formal analysis, Investigation, Methodology, Writing - original draft, Project administration; Cynthia Bailey, Data curation, Formal analysis, Investigation, Methodology; Meghana Mahantesh Magadam, Conceptualization, Data curation, Formal analysis, Investigation, Methodology; Shuyan Qiu, Data curation, Formal analysis; Karen L McNally, Data curation, Formal analysis, Investigation; Francis J McNally, Conceptualization, Data curation, Formal analysis, Supervision, Funding acquisition, Investigation, Writing - original draft, Project administration, Writing - review and editing

## Author ORCIDs
Francis J McNally https://orcid.org/0000-0003-2106-3062

Joint Public Review: https://doi.org/10.7554/eLife.97812.3.sa1
Author response https://doi.org/10.7554/eLife.97812.3.sa2

## Additional files

### Supplementary files
• Supplementary file 1. *C. elegans* strains used in this study.
• Source data 1. Excel spreadsheet of all numerical data values.
• MDAR checklist

### Data availability
All data generated or analysed during this study are included in the manuscript and supporting files.

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
