## [Editor Report · eLife assessment]

This is a **valuable** paper that identifies a potential challenge for embryos during fertilization: holding sperm contents in the fertilized embryos away from the oocyte meiotic spindle so that they don't get ejected into the polar body during meiotic chromosome segregation. The authors identify proteins involved in cytoplasmic streaming and maintaining the grouping of paternal organelles as being critical for this process. There remain minor weaknesses in the data presented but the paper provides **solid** evidence for the majority of its claims, and while the findings may pertain to a narrow audience the tools used and basic characterization shown will likely be relied upon by many in the community and therefore is of high value.

---

## [Referee Report · Joint Public Review]

Summary:

This paper by Beath et. al. identifies a potential regulatory role for proteins involved in cytoplasmic streaming and maintaining the grouping of paternal organelles: holding sperm contents in the fertilized embryos away from the oocyte meiotic spindle so that they don't get ejected into the polar body during meiotic chromosome segregation. The authors show that by time-lapse video, paternal mitochondria (used as a readout for sperm and its genome) is excluded from yolk granules and maternal mitochondria, even when moving long distances by cytoplasmic streaming. To understand how this exclusion is accomplished, they first show that it is independent of both internal packing and the engulfment of the paternal chromosomes by the maternal endoplasmic reticulum creating an impermeable barrier. They then test whether the control of cytoplasmic steaming affects this exclusion by knocking down two microtubule motors, Katanin and kinesis I. They find that the ER ring, which is used as a proxy for paternal chromosomes, undergoes extensive displacement with these treatments during anaphase I and interacts with the meiotic spindle, supporting their hypothesis that the exclusion of paternal chromosomes is regulated by cytoplasmic streaming. Next, they test whether a regulator of maternal ER organization, ATX-2, disrupts sperm organization so that they can combine the double depletion of ATX-2 and KLP-7, presumably because klp-7 RNAi (unlike mei-1 RNAi) does not affect polar body extrusion and they can report on what happens to paternal chromosomes. They find that the knockdown of both ATX-2 and KLP-7 produces a higher incidence of what appears to be the capture of paternal chromosomes by the meiotic spindle (5/24 vs 1/25). However, this capture event appears to halt the cell cycle, preventing the authors from directly observing whether this would result in the paternal chromosomes being ejected into the polar body.

The authors addressed the vast majority of the Reviewer's comments including the addition of new figures, re-wording of data interpretation and discussion points to better reflect the claims of the paper. There remain a few outstanding points which were not addressed.

In many cases the number of embryos analyzed or events capture remains low and the authors conclude that these sample sizes prevented statistical significance. It's not clear if more embryos were analyzed or if more capture would lead to statistical significance. Language capturing this caveat should also be included in the manuscript. A specific example of this is given below:

In the double knockdown of ATX-2 and KLP-7, there was no significant difference between single and double knockdowns and the ER ring displacement was not analyzed in this double mutant. Further, there was no difference in the frequency of sperm capture between single and double ATX-2 and KLP-7 due to low sample size, the the strength of the conclusion of this manuscript would be greatly improved if both of these results were further explored.

---

## [Author Response]

The following is the authors’ response to the original reviews.

Public Reviews:
**Reviewer #1 (Public Review):**
Summary:This paper by Beath et. al. identifies a potential regulatory role for proteins involved in cytoplasmic streaming and maintaining the grouping of paternal organelles: holding sperm contents in the fertilized embryos away from the oocyte meiotic spindle so that they don't get ejected into the polar body during meiotic chromosome segregation. The authors show that by time-lapse video, paternal mitochondria (used as a readout for sperm and its genome) is excluded from yolk granules and maternal mitochondria, even when moving long distances by cytoplasmic streaming. To understand how this exclusion is accomplished, they first show that it is independent of both internal packing and the engulfment of the paternal chromosomes by maternal endoplasmic reticulum creating an impermeable barrier. They then test whether the control of cytoplasmic steaming affects this exclusion by knocking down two microtubule motors, Katanin and kinesis I. They find that the ER ring, which is used as a proxy for paternal chromosomes, undergoes extensive displacement with these treatments during anaphase I and interacts with the meiotic spindle, supporting their hypothesis that the exclusion of paternal chromosomes is regulated by cytoplasmic streaming. Next, they test whether a regulator of maternal ER organization, ATX-2, disrupts sperm organization so that they can combine the double depletion of ATX-2 and KLP-7, presumably because klp-7 RNAi (unlike mei-1 RNAi) does not affect polar body extrusion and they can report on what happens to paternal chromosomes. They find that the knockdown of both ATX-2 and KLP-7 produces a higher incidence of what appears to be the capture of paternal chromosomes by the meiotic spindle (5/24 vs 1/25). However, this capture event appears to halt the cell cycle, preventing the authors from directly observing whether this would result in the paternal chromosomes being ejected into the polar body.Strengths:This is a useful, descriptive paper that highlights a potential challenge for embryos during fertilization: when fertilization results in the resumption of meiotic divisions, how are the paternal and maternal genomes kept apart so that the maternal genome can undergo chromosome segregation and polar body extrusion without endangering the paternal genome? In general, the experiments are well-executed and analyzed. In particular, the authors' use of multiple ways to knock down ATX-2 shows rigor.Weaknesses:The paper makes a case that this regulation may be important but the authors should do some additional work to make this case more convincing and accessible for those outside the field. In particular, some of the figures could include greater detail to support their conclusions, they could explain the rationale for some experiments better and they could perform some additional control experiments with their double depletion experiments to better support their interpretations. Also, the authors' inability to assess the functional biological consequences of the capture of the sperm genome by the oocyte spindle should be discussed, particularly in light of the cell cycle arrest that they observe.

These general comments are addressed in the more specific critiques below.

**Reviewer #2 (Public Review):**
SummaryIn this manuscript, Beath et al. use primarily *C. elegans* zygotes to test the overarching hypothesis that cytoplasmic mechanisms exit to prevent interaction between paternal chromosomes and the meiotic spindle, which are present in a shared zygotic cytoplasm after fertilization. Previous work, much of which by this group, had characterized cytoplasmic streaming in the zygote and the behavior of paternal components shortly after fertilization, primarily the clustering of paternal mitochondria and membranous organelles around the paternal chromosomes. This work set out to identify the molecular mechanisms responsible for that clustering and test the specific hypothesis that the "paternal cloud" helps prevent the association of paternal chromosomes with the meiotic spindle.StrengthsThis work is a collection of technical achievements. The data are primarily 3- and 4-channel time-lapse images of zygotes shortly after fertilization, which were performed inside intact animals. There are many instances in which the experiments show extreme technical skill, such as tracking the paternal chromosomes over large displacements throughout the volume of the embryo. The authors employ a wide variety of fluorescent reporters to provide a remarkably clear picture of what is going on in the zygote. These reagents and the novel characterization of these stages that they provide will be widely beneficial to the community.The data provide direct visualization of what had previously been a mostly hypothetical structure, the "paternal cloud," using simultaneous labeling of paternal DNA and mitochondria in combination with a variety of maternal proteins including maternal mitochondria, yolk granules, tubulin, and plasma membrane. Together, these images provided convincing evidence of the existence of this specified cytoplasmic domain. They go on to show that the knockdown of the ataxin-2 homolog ALX-2, a protein previously shown to affect ER dynamics, disrupted the paternal cloud, identifying a role for ER organization in this structure.The authors then used the system to test the functional consequences of perturbing the cytoplasmic organization. Consistent with the paternal cloud being a stable structure, it stayed intact during large movements the authors generated using previously published knockdowns (of mei-1/katanin and kinesin-13/kpl-7) that increased cytoplasmic streaming. They used this data to document instances in which the paternal chromosomes were likely to have been attached to the spindle. They concluded with direct evidence of spindle fibers connecting to the paternal chromatin upon knockdown of ATX-2 in combination with increased cytoplasmic streaming, providing strong, direct support for their overarching hypothesis.WeaknessesWhile the data is convincing, the narrative of the paper could be streamlined to highlight the novelty of the experiments and better articulate the aims. For example, the cloud of paternal mitochondria and membranous organelles was previously shown, but Figures 1-2 largely reiterate that observation. The innovation seems to be that the combination of ER, yolk, and maternal mitochondrial markers makes the existence of a specified domain more concrete. There are also some instances where more description is needed to make the conclusions from the images clear.

These general comments are addressed in the more specific critiques below.

The manuscript intersperses what read like basic characterizations of fluorescent markers that, as written, can distract from the main story. The authors characterized the dynamics of ER organization throughout the substages of meiosis and the permeability of the envelope of ER that surrounds the paternal chromatin, but it could be more clearly established how the ability to visualize these structures allowed them to address their aims.

We have added the following after the initial description of ER morphology changes: (ER morphology was used to determine cell-cycle stages during live imaging reported below in Fig. 6.)

More background on what was previously known about ER organization in M-phase and the role of ataxin proteins specifically may help provide more continuity.

We have added references to transitions to ER sheets during mitotic M-phase in HeLa cells and *Xenopus* extracts.

**Reviewer #3 (Public Review):**
Summary:This study by Beath et al. investigated the mechanisms by which sperm DNA is excluded from the meiotic spindle after fertilization. Time-lapse imaging revealed that sperm DNA is surrounded by paternal mitochondria and maternal ER that is permeable to proteins. By increasing cytoplasmic streaming using kinesin-13 or katanin RNAi, the authors demonstrated that limiting cytoplasmic streaming in the embryo is an important step that prevents the capture of sperm DNA by the oocyte meiotic spindle. Further experiments showed that the Ataxin-2 protein is required to hold paternal mitochondria together and close to the sperm DNA. Finally, double depletion of kinesin-13 and Ataxin-2 suggested an increased risk of meiotic spindle capture of sperm DNA.Overall, this is an interesting finding that could provide a new understanding of how meiotic spindle capture of sperm DNA and its accidental expulsion into the polar body is prevented. However, some conceptual gaps need to be addressed and further experiments and improved data analyses would strengthen the paper.- It would be helpful if the authors could discuss in good detail how they think maternal ER surrounds the sperm DNA

We have added 2 references to papers about nuclear envelope re-assembly from Shirin Bahmanyar’s lab and suggest the ER envelope is a halted intermediate in nuclear envelope reassembly.

and why is it not disrupted following Ataxin disruption.

We have been attempting to disrupt ER structures in the meiotic embryo for the last 5 years by depleting profilin, BiP, atlastin, ATX-2 and by optogenetically packing ER into a ball in the middle of the oocyte. None of these treatments prevent envelopment of the sperm DNA by maternal ER. None of these treatments remove ER from the spindle envelope and none remove ER from the plasma membrane. These treatments mostly result in “large aggregates” of ER that we have not examined by EM. Wild speculation: any disruption of the ER strong enough to prevent ER envelopment around chromatin would be sterile because the M to S transition in the mitotic zone of the germline would be blocked. Rapid depletion of ATX-2 to the extent shown by rigorous data in this manuscript does not prevent ER envelopment around chromatin. We chose not to speculate about the reasons for this because we do not know why.

- Since important phenotypes revealed in RNAi experiments (e.g. kinesin-13 and ataxin-2 double depletion) are not very robust, the authors should consider toning down their conclusions and revising some of their section headings. I appreciate that they are upfront about some limitations, but they do nonetheless make strong concluding sentences.

We have changed the discussion of the klp-7 atx-2 double depletion to: “The capture of the sperm DNA by the meiotic spindle in ATX-2 KLP-7 double depleted embryos suggests that the integrity of the exclusion zone around the sperm DNA might insulate the sperm DNA from spindle microtubules. However, a much larger number of *klp-7(RNAi)* singly depleted and *atx-2(degron)* singly depleted time-lapse sequences are needed to rigorously support this idea. “

- The discussion section could be improved further to present the authors' findings in the larger context of current knowledge in the field.

We have expanded the discussion as suggested.

- The authors previously demonstrated that F-actin prevents meiotic spindle capture of sperm DNA in this system. However, the current manuscript does not discuss how the katanin, kinesin-13 and Ataxin-2 mechanisms could work together with previously established functions of F-actin in this process.

We have added pfn-1(RNAi) to the discussion section.

- How can the authors exclude off-target effects in their RNAi depletion experiments? Can kinesin-13, katanin, and Ataxin phenotypes be rescued for instance?

For ataxin-2 phenotypes, two completely independent controls for off target effects are shown. GFP(RNAi) on a strain with and endogenous ATX-2::GFP tag vs GFP(RNAi) on a strain with no tag on the ATX-2. ATX-2::AID with or without auxin. For kinesin-13 and katanin, we did not do a rigorous control for off-target effects of RNAi. However, the effects of these depletions on cytoplasmic microtubules have been previously reported by others

- How are the authors able to determine if the paternal genome was actually captured by the spindle? Does lack of movement definitively suggest capture without using a spindle marker?

mKate::tubulin labels the spindle in each capture event. This can be seen in Video S3. for mei-1(RNAi) and Figure 9 for atx-2 klp-7 double depletions.

(1) Major issues:The images provided are not convincing that mitochondria are entirely excluded from the regions with yolk granules from the images provided. Please provide insets of magnified images of the paternal mitochondria in Figure 1E to more clearly show the exclusion even when paternal mitochondria are streaming. Providing grayscale images, individual z-sections and/or some quantification of this data might also be more convincing to this reviewer.

We have modified Fig. 1 by adding single wavelength magnified insets to more clearly show that paternal mitochondria are in a “black hole” in the maternal yolk granules during cytoplasmic streaming.

Figure 2 -This figure can be retitled to highlight that the paternal organelle cloud is impermeable to mitochondria and conserved.

The legend has been re-titled as suggested.

Figure 3B, An image of the DNA within the ring of maternal ER especially since the maternal ER ring is used as a proxy for the paternal chromosomes in later figures would strengthen the authors' claims.

We have added a panel showing DAPI-stained DNA in the center of the ER ring and paternal mitochondria cloud.

Why is the faster time scale imaging significant? I think this could be more clearly set up in the paper. Perhaps rapid imaging of maternal mito-labeled kca-1(RNAi) embryos would better show the difference in time scale, with the expectation that the paternal cloud forms and persists while the ER invades.

We are not sure what the reviewer means. 5 sec time intervals were used throughout the paper. We are also not sure how kca-1(RNAi) would help. Movement of the entire oocyte into and out of the spermatheca is what limits the ability to keep a fusing sperm in focus. kca-1(RNAi) would prevent cytoplasmic streaming but not ovulation movements.

Figure 4 - The question about the permeability of the ER envelope seems to come out of nowhere as written. It isn't clear how it contributes to the larger story about preventing sperm incorporation in the spindle.

This section of the results is introduced with: “If the maternal ER envelope around sperm DNA was sealed and impermeable during meiosis, this could both prevent the sperm DNA from inducing ectopic spindle assembly and prevent the sperm DNA from interacting with meiotic spindle microtubules.”

The data in Figure 4 would probably not be expected to be in this paper based on the paper title. Maybe the title needs something about ER dynamics? "eg. ATX-2 but not an ER envelope" isolates the paternal chromatin?In Figure 5, it seems that RNAi of klp-7 and Mei-1 had slightly different effects on short-axis displacement of the ER envelope (klp-7 affecting it more dramatically than mei-1) and slightly different effects on interaction with the meiotic spindle (capture vs streaming past the spindle). The authors mention in their discussion that the difference in the interaction with the meiotic spindle might reflect the effects that loss of Mei-1 may have on the spindle but could it also be a consequence of the differences in cytoplasmic streaming observed?

With our current data, the only statistically significant difference between cytoplasmic streaming of the sperm contents in mei-1(RNAi) vs klp-7(RNAi) is that excessive streaming persists longer into metaphase II in klp-7(RNAi). We have added a sentence describing this difference to the results. If differences in streaming were the cause of different capture frequencies, then klp-7(RNAi) would cause more capture events than mei-1(RNAi) but the opposite was observed. We have avoided too much discussion here because the frequency of capture events is too low to demonstrate statistically significant differences between mei-1(RNAi), klp-7(RNAi), and atx-2(degron) + klp-7(RNAi) without a very large increase in the number of time-lapse sequences.

Also, the authors should find a way to represent this interaction with the meiotic spindle in a quantitative or table form to allow the reader to observe some of the patterns they report more easily.

We have added a table to Fig. 9 that summarizes capture data.

Finally, can the authors report when they observe the closest association with the meiotic spindle: Does it correlate with the period of greatest displacement (AI) or are they unlinked?

The low frequency of capture events makes it difficult to test this rigorously.

Figure 6- 'Endogenously tagged ATX-2 was observed throughout oocytes and meiotic embryos without partial co-localization with ER.' How can the authors exclude co-localization with ER?

We have changed the wording to: “Endogenously tagged ATX-2 was observed throughout oocytes and meiotic embryos (Fig. 6A; Fig. S2). ATX-2 did not uniquely co-localize with ER (Fig. S2).“

The rationale for why the authors think that the integrity of sperm organelles is important to keep the genomes apart is not clear to this reviewer and needs to be explained better. Moving the discussion of the displacement experiments in Figure S3 from the end of the results section to the ATX-2 knockdown section would help accomplish this.

We have added the sentence: “The frequency of sperm capture by the meiotic spindle (Fig. 9D) was significantly higher than wild-type controls in *klp-7(RNAi) atx-2(AID)* double depleted embryos (p=0.011 Fisher’s exact test). Although the number of single mutant embryos analyzed was too low to demonstrate a significant difference between single and double mutant embryos, these results qualitatively support the hypothesis that limiting cytoplasmic streaming and maintaining the integrity of the ball of paternal mitochondria are both important for preventing capture events between the meiotic spindle and sperm DNA.”

It looks like, in the double knockdown of ATX-2 and KLP-7, the spread of paternal mitochondria is less affected than when only ATX-2 is depleted. What effect does this result have on the observation that the incidence of sperm capture appears to increase in the double depletion? What does displacement of the ER ring look like in the double depletion? Is it additive, consistent with their interpretation that both limiting cytoplasmic streaming and maintaining the integrity of the ball of paternal mitochondria is required to keep the genomes separate?

We cannot show a significant difference between single a double knockdowns without increasing n by alot. We did not analyze ER ring displacement in the double mutant.

Is the increased incidence of capture in the double-depleted embryos significant?

We have added the sentence: “The frequency of sperm capture by the meiotic spindle (Fig. 9D) was significantly higher than wild-type controls in *klp-7(RNAi) atx-2(AID)* double depleted embryos (p=0.011 Fisher’s exact test). Although the number of single mutant embryos analyzed was too low to demonstrate a significant difference between single and double mutant embryos, these results qualitatively support the hypothesis that limiting cytoplasmic streaming and maintaining the integrity of the ball of paternal mitochondria are both important for preventing capture events between the meiotic spindle and sperm DNA.”

What do the authors make of the cell cycle arrest observed when paternal chromosomes are captured? Is there an argument to be made that this arrest supports the idea that preventing this capture is actively regulated and therefore functionally important?

We chose not to discuss the mechanism of this arrest because considerably more work would be required to prove that it is not caused by a combination of imaging conditions and genotype. The low frequency of these capture + arrest events would make it very difficult to show that the arrest does not occur after depleting a checkpoint protein.

(2) Minor concerns:Top of page 4: "streaming because depletion tubulin stops cytoplasmic streaming (7)" should be "streaming because depletion of tubulin stops cytoplasmic streaming (7)"

The ”of” has been inserted.

Page 6: "This result indicated that the volume of paternal mitochondria excludes maternal mitochondria and yolk granules but not maternal ER." The authors have only shown this for maternal mitochondria, not yolk granules.

We have deleted the mention of yolk granules here.

Page 7: "These results suggest that all maternal membranes are initially excluded from the sperm at fusion." Should be "These results show that maternal ER are initially excluded from the sperm at fusion. Since maternal mitochondria and yolk granules are excluded later, this suggests that all maternal membranes are initially excluded from the sperm at fusion."

We have changed this sentence as suggested.

It's not clear why the authors show other types of movement that might be quantified when cytoplasmic streaming is affected in Figure 5A and only quantify long-axis and short-axis displacement.

We have deleted the other types of movement from the schematic. Although these parameters were quantified, we did not include this data in the results so it would be confusing for the reader to have them in the schematic.

Bottom of page 7: Mention that the GFP::BAF-1 was maternally provided.

We have added “Maternally provided..”

Missing an Arrow on Figure 1A 9:20.

We removed the text citation to an arrow in Fig. 1A because we moved most of the description of the ER ring to Fig. 3 to address other reviewer suggestions.

Supplemental videos should be labeled appropriately to indicate what structures are labeled. It is currently difficult to understand what is being shown.(3) Issues with the Discussion section:"The simplest explanation is that cytoplasm does not mix during the 45 min from GVBD to pronucleus formation due to the high viscosity of cytoplasm." - Citation page 12.

We have changed the sentence to: “The simplest hypothesis is that maternal and paternal cytoplasm might not mix during the 45 min from GVBD to pronucleus formation due to the high viscosity of cytoplasm.”

"The higher frequency of capture of the sperm DNA by the meiotic spindle in ATX-2 KLP-7 double depleted embryos compared with either single depletion suggests that the integrity of the exclusion zone around the sperm DNA may insulate the sperm DNA from spindle microtubule" - Pages 12-13 reference the figures.

This sentence has been rewritten in response to other comments but the new sentence now references revised Fig. 9.

"ATX-2 is required to maintain the integrity of the ball of paternal mitochondria around the sperm DNA, but the mechanism is unknown." - Page 13 reference figure.

A reference to Figs 7 and 8 has been inserted.

" In control embryos, the sperm contents rarely came near the meiotic spindle in agreement with a previous study that found that male and female pronuclei rarely form next to each other (6). Streaming of the sperm contents was most commonly restricted to a jostling motion with little net displacement, circular streaming in the short axis of the embryo, or long axis streaming in which the sperm turned away from the spindle before the halfway point of the embryo. Depletion of MEI-1 or KLP-7 resulted in longer excursions of the sperm contents in the long axis of the embryo toward the spindle but frequent capture of the sperm by the spindle was only observed in mei-1(RNAi)." - Page 13, the corresponding figures need to be referenced for these sentences.

We have inserted figure references.

"In capture events observed after double depletion of ATX-2 and KLP-7, a bundle of microtubules was discernible extending from the spindle into the ER envelope surrounding the sperm DNA. Such bundles were not observed in mei-1(RNAi) capture events, likely because of the previously reported low density of microtubules in mei-1(RNAi) spindles (36, 37)." - Pages 13-14 references figures here.

We have inserted figure references.

"The higher frequency of capture of the sperm DNA by the meiotic spindle in ATX-2 KLP-7 double depleted embryos compared with either single depletion suggests that the integrity of the exclusion zone around the sperm DNA may insulate the sperm DNA from spindle microtubules." - This should be toned down since this phenotype is not robust.

We have changed this to: “The capture of the sperm DNA by the meiotic spindle in ATX-2 KLP-7 double depleted embryos suggests that the integrity of the exclusion zone around the sperm DNA might insulate the sperm DNA from spindle microtubules. However, a much larger number of *klp-7(RNAi)* singly depleted and *atx-2(degron)* singly depleted time-lapse sequences are needed to rigorously support this idea. “

ATX-2 depletion alters ER morphology but does not impact the maternal ER envelope - could the authors provide a potential explanation for this?

In the discussion, we cite papers showing that ATX-2 depletion affects many different cellular processes so the effect we see on paternal mitochondria might have nothing to do with the ER ring. We have been attempting to disrupt ER structures in the meiotic embryo for the last 5 years by depleting profilin, BiP, atlastin, ATX-2 and by optogenetically packing ER into a ball in the middle of the oocyte. None of these treatments prevent envelopment of the sperm DNA by maternal ER. None of these treatments remove ER from the spindle envelope and none remove ER from the plasma membrane. These treatments mostly result in “large aggregates” of ER that we have not examined by EM. Wild speculation: any disruption of the ER strong enough to prevent ER envelopment around chromatin would be sterile because the M to S transition in the mitotic zone of the germline would be blocked. Rapid depletion of ATX-2 to the extent shown by rigorous data in this manuscript does not prevent ER envelopment around chromatin. We chose not to speculate about the reasons for this because we do not know why.

It would be good to have representative images of what the altered spindle looks like in MEI-1-depleted oocytes.

The structure of MEI-1-depleted spindles has been described in the cited references.

"Depletion of MEI-1 or KLP-7 resulted in longer excursions of the sperm contents in the long axis of the embryo toward the spindle but frequent capture of the sperm by the spindle was only observed in mei-1(RNAi)" - It is intriguing that this does not happen in the double depletion experiments of kinesin-13 and ATX-2. The authors should perhaps discuss this.

This does happen in KLP-7 ATX-2 double depleted embryos as shown in Fig. 9.

(4) Missing citations:"This analysis was restricted to embryos from anaphase I through anaphase II because our streaming data and that of Kimura 2020 indicate that the sperm contents have not moved significantly before anaphase I." - This needs an appropriate citation. Page 10.

We have inserted citations here.

" The simplest explanation is that cytoplasm does not mix during the 45 min from GVBD to pronucleus formation due to the high viscosity of cytoplasm." - Citation page 12. Not referencing figures in the discussion.

We have changed the sentence to: “The simplest hypothesis is that maternal and paternal cytoplasm might not mix during the 45 min from GVBD to pronucleus formation due to the high viscosity of cytoplasm.”

"The higher frequency of capture of the sperm DNA by the meiotic spindle in ATX-2 KLP-7 double depleted embryos compared with either single depletion suggests that the integrity of the exclusion zone around the sperm DNA may insulate the sperm DNA from spindle microtubule" - Pages 12-13 reference the figures.

A reference to the revised Fig. 9 has been inserted in the revised version of this sentence.

"ATX-2 is required to maintain the integrity of the ball of paternal mitochondria around the sperm DNA, but the mechanism is unknown."

References to Figs. 7 and 8 have been inserted.

Page 13 reference figure" In control embryos, the sperm contents rarely came near the meiotic spindle in agreement with a previous study that found that male and female pronuclei rarely form next to each other (6). Streaming of the sperm contents was most commonly restricted to a jostling motion with little net displacement, circular streaming in the short axis of the embryo, or long axis streaming in which the sperm turned away from the spindle before the halfway point of the embryo. Depletion of MEI-1 or KLP-7 resulted in longer excursions of the sperm contents in the long axis of the embryo toward the spindle but frequent capture of the sperm by the spindle was only observed in mei-1(RNAi)." Page 13, the corresponding figures need to be referenced for these sentences.

We have inserted citations here.

"In capture events observed after double depletion of ATX-2 and KLP-7, a bundle of microtubules was discernible extending from the spindle into the ER envelope surrounding the sperm DNA. Such bundles were not observed in mei-1(RNAi) capture events, likely because of the previously reported low density of microtubules in mei-1(RNAi) spindles (36, 37)." Pages 13-14 references figures here.

We have inserted citations here.

(5) Referencing wrong figures in the text:Figure 5 - In the figure legend there is a 5C but there is no 5C panel in the figure.

A C has been inserted in Fig. 5.

Figure 6A - "Dark holes were observed suggesting exclusion from the lumens of larger membranous organelles (Fig. 6A; Fig. S2)." Page 10.

6A has been changed to 6C.

Figure 6A is showing background autofluorescence in WT oocytes so I am not certain why it is cited here.

The Figure citation has been corrected to 6B, C.

Figure 8 - I could not find the supplemental data file with the individual mitochondria distance measurements.

We are including the Excel file with the revised submission.

The last sentence of the first paragraph should be re-worded to be more concise ". In *C. elegans*, the nucleus is positioned away from the site of future fertilization so that the meiosis I spindle assembles at the opposite end of the ellipsoid zygote from the site of fertilization (2-4). "

Every word of this sentence is important.

Last sentence second paragraph typo "These microtubules are thought to drive meiotic cytoplasmic streaming because depletion tubulin stops cytoplasmic streaming (7) and depletion of the microtubule-severing protein katanin by RNAi results in an increased mass of cortical microtubules and an increase in cytoplasmic streaming (8)." Pages 3-4.

“of” has been inserted.

(6) Typos in the introduction should be corrected:Ataxin or kinesin-13 are not mentioned in the introduction but these are a big focus of the paper.Gong et al 2024 written instead of number citation (page 5), no citation in References.

This has been corrected.

Supplemental videos should be labeled appropriately to indicate what structures are labeled. It is currently difficult to understand what is being shown.